# AReaL-DTA: Dynamic Tree Attention
# for Efficient Reinforcement Learning of Large Language Models

**Jiarui Zhang** [* 1 2]  **Yuchen Yang** [* 2]  **Ran Yan** [* 1 3]  **Zhiyu Mei** [3]  **Liyuan Zhang** [1]  **Daifeng Li** [1]  **Wei Fu** [2 3]
**Jiaxuan Gao** [2 3]  **Shusheng Xu** [3]  **Yi Wu** [2]  **Binhang Yuan** [1]

## Abstract

Reinforcement learning (RL)-based post-training for large language models (LLMs) is computationally expensive, as it generates many rollout sequences that frequently share long token prefixes. Existing RL frameworks usually process these sequences independently during policy training, i.e., repeatedly recomputing identical prefixes in both the forward and backward passes of policy gradient computation, leading to substantial inefficiencies in computation resources and memory usage. Although prefix sharing naturally induces a tree structure over rollouts, packed tree-mask approaches scale poorly in RL settings. In this paper, we introduce AReaL-DTA, which efficiently exploits prefix sharing in RL training. AReaL-DTA employs a depth-first search (DFS)-based execution strategy that dynamically traverses the rollout prefix tree during both forward and backward computation, materializing only a single root-to-leaf path at a time. To further improve scalability, AReaL-DTA incorporates a load-balanced distributed batching mechanism that dynamically constructs and processes prefix trees across multiple GPUs. On $\tau^2$-bench, AReaL-DTA improves training throughput by up to $8.31\times$ over dense training and up to $1.70\times$ over sparse training. Our code is available at https://github.com/areal-project/AReaL/tree/feat/dta.

[*]Equal contribution  [1]The Hong Kong University of Science and Technology, Hong Kong SAR, China [2]Tsinghua University, Beijing, China [3]AReaL Team, Ant Group, Hangzhou, China. Correspondence to: Binhang Yuan <biyuan@ust.hk>.

*Proceedings of the 43$^{rd}$ International Conference on Machine Learning*, Seoul, South Korea. PMLR 306, 2026. Copyright 2026 by the author(s).

## 1. Introduction

Post-training large language models (LLMs) with reinforcement learning (RL) is often computationally intensive (Hu et al., 2025; Fu et al., 2025). RL post-training workflows commonly generate many rollout sequences for each prompt or environment state to explore different outcomes or gather reward signals (Hendrycks et al., 2021; Wang et al., 2025b). These rollouts frequently share long prefixes, such as the same prompt, system instruction, or earlier dialogue turns (Wang et al., 2025a). However, current policy-training pipelines usually process each rollout as an independent sequence, repeatedly recomputing the same prefix tokens in every forward and backward pass. This redundant computation wastes GPU time and memory, making policy-model training a major throughput bottleneck.

Prefix sharing is ubiquitous in modern RL training workflows for LLMs. For instance, RL for advanced reasoning LLMs and multi-turn agent training often sample multiple continuations per context, all starting from shared prompts, dialogue histories, or environment states (Chen et al., 2021; Hendrycks et al., 2021; Wang et al., 2025b; Gao et al., 2025). These branching trajectories naturally form a prefix tree: they share a common stem before diverging into different outcomes. We characterize the resulting reuse opportunity by the compression rate $C = \eta_{\text{tokens}}/\eta_{\text{tree tokens}}$, i.e., the ratio between total sequence tokens and unique prefix-tree tokens, and the corresponding prefix-sharing rate $S = 1 - 1/C$. As illustrated by the $\tau^2$-bench (Yao et al., 2024) example in Figure 1, efficiently reusing shared prefixes can eliminate a large fraction of duplicate work, yielding proportionate savings in runtime and enabling larger batch sizes or more samples without exhausting memory.

However, exploiting prefix sharing in policy model training is challenging due to the attention mechanism in the transformer architecture and the complexity of the prefix tree branching computations. One natural sparse solution, which we refer to as sparse, packs the rollout prefix tree into one masked forward/backward computation, using a tree attention mask to ensure that each token attends only to its valid prefix context (Cai et al., 2024; Miao et al., 2024). Such an approach avoids recomputing shared prefixes, but

## Shared Prefix Tree Structure for $\tau^2$-bench

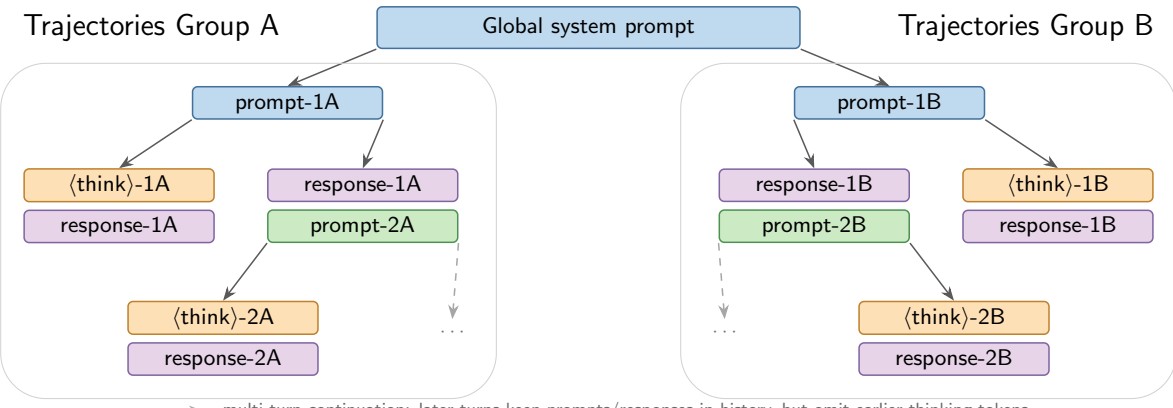

*Figure 1.* Shared-prefix structure in $\tau^2$-bench. Multi-turn trajectories share a global system prompt and retain previous prompts/responses in later turns, naturally inducing a prefix tree over rollout sequences. We define the compression rate as $C = \eta_{\text{tokens}}/\eta_{\text{tree tokens}}$, i.e., total sequence tokens divided by unique prefix-tree tokens. In our evaluated rollouts, the full tree has $C = 9.43\times$, equivalent to an $89.4\%$ prefix-sharing rate ($S = 1 - 1/C$). Even after collapsing rollouts that are prefixes of other rollouts and counting only leaf sequences, the leaf-only compression rate remains $5.56\times$ ($82.0\%$ prefix sharing).

it still scales poorly in RL post-training workflows for two reasons. First, because Sparse executes the packed tree as a single training workload, its training-state memory still grows with the number of packed tree tokens, including activations, key/value tensors, and backward intermediates. Large trees can therefore require activation recomputation and splitting into smaller subtrees to fit GPU memory, which lowers effective prefix reuse and reintroduces duplicated prefix computation across subtrees. Second, tree-mask execution relies on specialized sparse attention kernels, whose irregular access patterns can achieve lower model FLOPs utilization (MFU) than highly optimized dense attention kernels (i.e., a causal mask). In practice, these memory and execution overheads often negate the benefit of prefix reuse.

In this paper, we introduce AREAL-DTA (i.e., **D**ynamic **T**ree **A**ttention over AReaL (Fu et al., 2025) RL training framework), a system that harnesses prefix sharing while addressing the inherited scaling challenges. The key idea is a depth-first search (DFS)-based dynamic computation strategy for forward and backward passes in Transformer-based policy model training. AREAL-DTA also includes a load-balanced distributed batching strategy that scales out RL training computation. Concretely, our contributions are enumerated as follows:

**Contribution 1**: We design and implement a *DFS traversal method over the token prefix tree*. Rather than packing the entire prefix tree into a single sparse masked computation, AREAL-DTA organizes rollouts as a prefix tree and visits one root-to-leaf path at a time. Shared-prefix segments are

backpropagated once and reused by descendant rollouts, while obsolete suffixes are backpropagated and released before the traversal moves to the next branch. This execution accumulates gradient contributions from shared prefixes correctly while keeping live activations proportional to the longest sequence depth rather than the total number of tokens in the prefix tree.

**Contribution 2**: To further scale up RL training, we develop a *load-balanced distributed batching strategy* for AREAL-DTA. Concretely, AREAL-DTA dynamically batches generated rollouts during an asynchronous rollout generation phase to construct multiple prefix trees and distributes computation across multiple training GPU workers, thereby balancing the computation load among trainer GPUs. This mechanism reduces GPU idle time and enables scalable prefix-tree construction and traversal for RL rollouts, even with many sequences and long trajectories in each policy model training iteration.

**Contribution 3**: We demonstrate substantial speed and memory gains on prefix-sharing-heavy RL training workloads. On $\tau^2$-bench, AREAL-DTA achieves up to $8.31\times$ throughput improvement for a single training worker, $6.20\times$ for the training cluster, and $2.28\times$ for the complete end-to-end pipeline. Compared with the dense baseline, AREAL-DTA also reduces peak GPU memory by over $50\%$ in our main ablation, reducing reliance on costly memory-saving techniques such as activation recomputation.

## 2. Preliminaries and Related Work

### 2.1. RL System for LLM Post-training

RL has been widely adopted to improve the reasoning abilities of large language models (LLMs) (OpenAI, 2024; 2025). Prior studies show that RL-based post-training can substantially boost performance on a broad range of reasoning-intensive tasks, including mathematical reasoning, program synthesis, and multi-hop question answering (Chen et al., 2021; Hendrycks et al., 2021; Wang et al., 2025b; Gao et al., 2025). From a systems perspective, RL training for LLMs is extremely resource-demanding and typically consists of three distinct stages: (**i**) *rollout generation*, which performs inference on GPUs limited by HBM I/O while producing multiple candidate responses (rollouts) for each prompt; (**ii**) *reward estimation*, which may rely on intensive CPU resources (e.g., sandboxed execution for code evaluation or rule-based solvers for mathematics) or additional GPU resources when LLM-based reward or value models are used; and (**iii**) *model training*, which carries out compute-intensive GPU updates of the policy (and optionally value) models via stochastic gradient optimization, sometimes involving a reference model for stabilization.

Existing RL training pipelines for LLMs generally fall into either synchronous or asynchronous paradigms. In *synchronous* RL training, rollout generation and model optimization are executed in alternating iterations: the current policy model parameters are first used to generate reasoning trajectories, and the resulting rollouts are then consumed to update the model (Shao et al., 2024; Sheng et al., 2025b; Qin et al., 2025; Hu et al., 2025). In contrast, *asynchronous* RL training allows these stages to proceed concurrently, where rollout generation continuously produces trajectories using possibly stale parameters while the training process updates the model in parallel (Mnih et al., 2016; Espeholt et al., 2018; 2019; Mei et al., 2023; Fu et al., 2025; Sheng et al., 2025a; Li et al., 2026). Among those asynchronous systems, AReaL (Fu et al., 2025) further decouples streaming generation from training through a fully asynchronous architecture, and introduces algorithmic techniques such as staleness-aware optimization and a decoupled RL objective to achieve efficient and stable RL training for LLM reasoning workflows.

### 2.2. Tree Attention for LLM Inference and Training

Organizing multiple sequences as a prefix tree was first explored to accelerate LLM inference by parallelizing speculative decoding and reusing shared prefixes across candidate outputs (Sun et al., 2023). For example, Specinfer (Miao et al., 2024) organizes candidate tokens into a token tree and verifies them in parallel with a single model pass, enabling the model to verify more tokens per iteration. Building on a similar perspective, Medusa (Cai et al., 2024) augments the original LLM with extra decoding heads to predict multiple subsequent tokens in one step and employs a tree-structured attention mask to construct and validate several continuation branches simultaneously at each step. Beyond these draft–verify frameworks, recent research has also optimized the tree decoding graph itself for greater efficiency. For example, Sequoia (Chen et al., 2024) applies a search-based strategy to allocate a fixed token budget across the tree's depth and width, maximizing prefix reuse under a given cost constraint. Yggdrasil (Guan et al., 2025) bridges dynamic speculation with static runtime optimization, dynamically selecting the tree's width (i.e., parallel branches) and depth for each query while using an "equal-growth" tree structure and staged scheduling to maintain high hardware utilization. Complementary optimization also targets the memory and compute overhead of tree-based attention (Yao et al., 2025; Pan et al., 2025). For example, FastTree (Pan et al., 2025) introduces specialized attention kernels that pack and tile computations for branches sharing common prefixes, reducing redundant key/value loads and memory access. Overall, reusing shared prefix states and exploring multiple token continuations through parallel decoding heads or branches, combined with optimized execution through custom attention masks, kernels, and scheduling, can reduce runtime and memory overhead.

In contrast to the widespread utilization in LLM inference, tree-based attention for LLM training has been relatively unexplored — the closest work is Tree Training (Wang et al., 2025a), which also targets RL fine-tuning by reusing prefix computations across branching trajectories. Tree Training packs trajectories into a shared prefix tree and uses a gradient-correction mechanism to preserve the original branch-wise training objective. This is closely related to the Sparse formulation studied in our evaluation: both materialize the packed prefix tree and execute an explicit tree-structured attention pattern. While this reduces duplicated prefix computation, packed-tree execution still keeps training states for the packed tree in GPU memory; when a large tree must be split into smaller subtrees due to memory constraints, the effective prefix reuse also drops. Moreover, its custom-kernel prefix-packing approach is not easily applicable without effective parallel training support. In contrast, AREAL-DTA uses dynamic DFS traversal and load-balanced distributed scheduling to address these essential challenges in RL training.

## 3. Dynamic Tree Attention

**Problem formulation**: We formalize policy-model training in RL as follows. Let $\mathbf{s}_1, \mathbf{s}_2, \ldots, \mathbf{s}_N$ be the set of $N$ rollout sequences used in the current policy model training iteration. Each sequence $\mathbf{s}_i$ has an associated loss signal $\mathcal{L}(\mathbf{s}_i)$, e.g., negative log-likelihood or an RL policy gradient signal, and

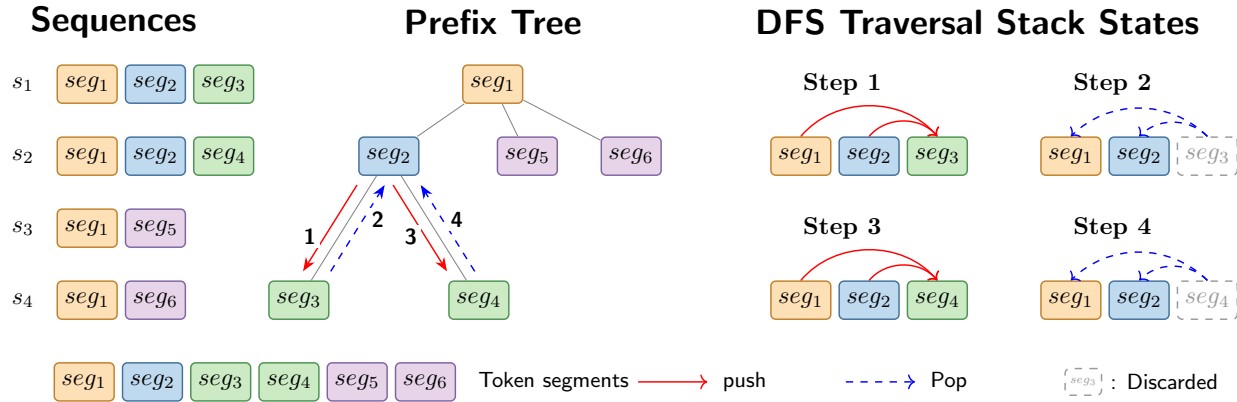

*Figure 2.* Illustration of the vanilla DTA push-pop execution. AREAL-DTA organizes rollout sequences ($s_1$–$s_4$) as a prefix tree and keeps only the active root-to-leaf path on a stack. Shared-prefix segments (e.g., $seg_1$ and $seg_2$) are pushed once and reused by multiple leaves; after a leaf loss is attached and its gradients are propagated, leaf-only segments (e.g., $seg_3$) are popped and released. The optimized execution further refines this vanilla DFS view with a backward DFS order and chunked pop operations; Algorithms 2 and 3 provide the corresponding pseudocode.

the total training objective is defined as:

$$\mathcal{L} = \sum_{i=1}^{N} \mathcal{L}(\mathbf{s}_i) \qquad (1)$$

To leverage the shared prefixes in these rollouts, we construct a prefix tree denoted by $\mathcal{T}$ that compactly represents all sequences and their shared prefixes. Each *node* in $\mathcal{T}$ corresponds to a segment of tokens shared by some subset of the sequences, and each root-to-leaf path corresponds to one full sequence, i.e., $\mathbf{s}_i$. By traversing this prefix tree, we can reuse computations for common prefixes among different rollouts.

The key challenge is achieving this reuse without introducing significant memory overhead or compromising the correctness of gradient computation. Vanilla AREAL-DTA addresses this challenge by dynamically traversing the prefix tree of all sequences in a depth-first manner and interleaving forward and backward passes to reuse computations (§3.1). We also conduct a series of system optimizations to improve computation efficiency and reduce memory overhead (§3.2). We provide the compact pseudocode in Appendix A.

### 3.1. Depth-First Search Traversal of Prefix Trees

The essential design of AREAL-DTA is to maintain a stack that represents the current path (prefix) from the root of the prefix tree to the current node we are visiting; Algorithm 2 provides the corresponding pseudocode. At any time, this stack holds (**i**) the sequence of tokens in the current prefix and (**ii**) the transformer's key/value cache for these prefix tokens. Given the prefix tree $\mathcal{T}$, the training computation (i.e., forward and backward propagation to compute the

policy gradient) proceeds as follows. In this vanilla DFS traversal, any prefix common to multiple sequences can be processed once in forward propagation, while gradients from all descendant sequences are accumulated across the corresponding backward steps instead of recomputing the prefix independently for each sequence.

- **Push prefix-tree nodes**: During DFS, moving from a prefix node to one of its children extends the current stack with the child token segment. AREAL-DTA runs a forward pass for this segment using the parent prefix's cached KV state, then appends the resulting KV cache to the stack. This reuses shared prefix states across descendant rollouts instead of recomputing them for each sequence.

- **Visit prefix tree leaf nodes**: As illustrated in Figure 2, when the DFS traversal reaches a leaf node corresponding to a full sequence $\mathbf{s}_i$, the stack now contains the complete token sequence $\mathbf{s}_i$ along with its forward pass results (e.g., log-probabilities, entropy). At this point, we can compute the loss $\mathcal{L}(\mathbf{s}_i)$ (i.e., by summing the negative log-likelihood of the correct tokens or applying the RL reward on the trajectory) using the information in the stack. We then immediately inject the loss gradient at the output of this sequence — effectively starting the backward pass for branch $\mathbf{s}_i$. By doing this as soon as a sequence's forward pass is done (rather than after processing all sequences), we ensure that the computation graph for that tail token segment does not need to remain in memory once its gradients are backpropagated.

- **Pop prefix tree intermediate nodes**: When DFS finishes all descendant branches of an intermediate node, the token segment represented by this node will no longer be used.

At this point, all gradient contributions to this segment's KV cache have been accumulated from its descendants, so AREAL-DTA backpropagates through the corresponding policy-model computations and pops the segment from the stack.

**Space complexity analysis**: Using this vanilla DFS traversal mechanism, AREAL-DTA keeps only the KV caches and activations for the current root-to-leaf path, plus a small amount of stack state. It therefore does not need to hold activations for the entire prefix tree simultaneously. The peak memory usage is proportional to the *length of the longest sequence* (i.e., the longest path in the prefix tree[1]) rather than to the total number of tree tokens across all sequences. In popular RL workflows where hundreds of sequences are processed together, this is a critical improvement: AREAL-DTA avoids the total-tree memory growth incurred by packed tree attention.

### 3.2. System Optimizations for AREAL-DTA

Note that we implement a series of system optimizations for the DFS traversal to further reduce memory usage and improve computation efficiency.

**Optimized AREAL-DTA execution**: Algorithm 2 presents a vanilla realization of dynamic tree attention: the traversal follows the tree recursively, performs a FWD pass when pushing a child segment, and immediately performs the corresponding BWD pass when returning from that child. Although this push-pop execution already reuses shared prefixes, it still couples every visited edge with one FWD and one BWD operation. In a rollout tree with $N_{\text{leaf}}$ leaves, this leads to roughly $2N_{\text{leaf}}$ FWD/BWD rounds along the leaf-to-leaf transitions, because many short token segments are pushed and popped separately. Algorithm 3 reorganizes the same computation around a leaf order $\pi$: for each leaf trajectory, AREAL-DTA performs one regular FWD pass to extend from the longest common prefix with the previous leaf, attaches the loss, and then pops the obsolete suffix. As a result, the main training computation is reduced to $N_{\text{leaf}}$ regular FWD/BWD rounds plus at most $N_{\text{leaf}}$ extra FWD-NOGRAD passes that materialize detached KV-cache anchors for later pop operations. These extra FWD-NOGRAD passes do not process every leaf's token segment; they only prepare cache anchors for subsequent computation graph construction and are triggered only when $\ell_{\text{lcp}} < \ell_{\text{next\_anchor}}$, so their runtime impact is limited. Moreover, for the $\tau^2$-bench rollout structure in Figure 1, the optimized DFS order ensures that each trajectory group typically requires only one such extra FWD-NOGRAD pass. The amount of extra FWD-NOGRAD work can be affected

by the chunk block size described below: larger blocks reduce this overhead, while smaller blocks increase it. In our profiled runs, this overhead accounts for around 7% of the total training time while keeping memory usage in check.

**Determining an optimal DFS traversal order**: Note that the order of DFS traversal over the prefix tree can substantially impact overall system efficiency. In AREAL-DTA, we design a *greedy algorithm* to optimize the DFS sequence. The greedy heuristic is simple yet effective: (**i**) minimize the number of extra forward passes by maximizing reuse of shared prefixes; and (**ii**) balance the backward pass segment lengths to avoid frequent short backward steps, which can become memory-bound. By prioritizing branches that lead to fewer extra forward passes and balancing gradient accumulation more evenly, our greedy algorithm improves memory efficiency and runtime stability across varying rollout tree structures. Algorithm 1 summarizes how this DFS order is used by the push-pop execution.

**Chunked backpropagation for long rollout suffixes**: We implement chunked backpropagation for long suffixes in the prefix tree, inspired by the chunk scheduling algorithm in ChunkFlow (Yuan et al., 2025). Although AREAL-DTA limits the live computation graph to a single suffix, that suffix can still span tens of thousands of tokens, making the corresponding FWD/BWD graph too large to fit in memory. To handle extremely long rollout trajectories, AREAL-DTA performs backpropagation over the suffix in fixed-length chunks. Given a maximum chunk length, e.g., 2048 tokens, AREAL-DTA splits a long suffix into consecutive chunks and performs chunked backpropagation from right to left. For each chunk, AREAL-DTA uses the stored prefix KV cache and outputs to recompute a local FWD pass, constructs the computation graph only for that chunk, immediately backpropagates through it, and then frees the associated activations before moving to the preceding chunk. This process is repeated until the entire suffix has been processed.

## 4. Load-Balanced Parallelization

While the dynamic tree attention algorithm above optimizes single-worker efficiency, large-scale RL training also demands distributed execution across multiple GPUs. In asynchronous RL frameworks, i.e., AREAL, rollouts are continuously generated and will be dispatched to multiple trainer GPUs for parallel training. For each policy model training iteration, AREAL-DTA leverages a load-balanced dispatcher that assigns incoming rollouts to different GPUs such that each GPU performs well-balanced work, maximizing overall throughput and avoiding idling.

**Problem formulation**: We formalize the dispatch problem as follows. Suppose at each policy model training iteration, we collect $N$ new rollouts from the rollout inference worker

---

[1] The path length is computed as the sum of token counts across the token segments on that path.

to train the policy model. We want to divide these $N$ sequences into $K$ disjoint groups (where $K$ is the number of trainer GPUs), and have each GPU build and process a prefix tree, denoted by $\mathcal{T}_j, j = 1, 2, \ldots, K$, for the sequences in its group. The cost of a group, denoted by $\mathcal{C}(\mathcal{T}_j)$, is defined by the estimated time to process its sequences as a prefix tree. Our goal is to partition the $N$ sequences into $K$ groups such that the maximum cost among the groups is minimized:

$$\mathcal{C} = \min \max_{j=1}^{K} \mathcal{C}(\mathcal{T}_j) \qquad (2)$$

In practice, we define $\mathcal{C}(\mathcal{T}_i)$ as the total number of tree tokens, obtained by summing the token segment lengths across all nodes in $\mathcal{T}_i$.

This optimization goal discourages overloading one GPU while others are under-utilized, i.e., it aims for all GPUs to finish their work in roughly the same time and achieve good scaling. The unrestricted version of this problem can be seen as a variant of balanced partitioning and is generally NP-hard. We therefore solve a DFS-contiguous ordered partitioning problem, which preserves most prefix sharing and admits an efficient optimal solution under the fixed order.

**Load-balanced partitioning algorithm**: We leverage the property of the prefix tree and a monotonic cost model to move toward a balanced partition efficiently. First, we arrange the $N$ sequences in a single prefix tree (as if we were to process them on one GPU). For this, we adopt lexicographical ordering, which is a valid DFS order for the prefix tree. In practice, we consider two types of overhead when splitting sequences across GPUs: (**i**) load imbalance among sequence groups; and (**ii**) duplication of prefix computations across groups (i.e., reduced prefix sharing relative to the combined prefix tree). A structure-agnostic partitioning that balances sequences purely by token count can drastically increase overhead (**ii**) by splitting shared prefixes across different GPUs. In contrast, sorting sequences by a DFS traversal order (which clusters sequences with common prefixes adjacently) and then partitioning this ordered list into $K$ contiguous segments reduces duplication. Intuitively, contiguous DFS segments preserve most prefix sharing: splitting into $K$ segments introduces at most $(K-1) \times \max(\text{len}(\mathbf{s}_i))$ additional tokens beyond the single combined prefix tree (where $\max(\text{len}(\mathbf{s}_i))$ is the length of the longest sequence), which is small relative to the total token count in our target workloads. Thus, we can reduce the prefix tree partitioning problem to a simpler sequence partitioning problem with limited loss of prefix reuse. Even with this DFS order constraint, sequences can be divided into nearly equal-sized contiguous blocks, maintaining balanced workloads across all GPUs.

Given this, we find the boundaries between the $K$ segments in the ordered list that minimize the maximum cost under the fixed contiguous-order constraint. We solve this via a binary search on the maximum allowed cost with a greedy check, a common approach for ordered partitioning problems. Specifically, we binary search for the smallest threshold $\tau$ such that we can cut the sequence list into at most $K$ segments, each with cost $\leq \tau$. For a candidate $\tau$, we scan through the sequences in order and greedily start a new segment whenever adding the next sequence would exceed the cost $\tau$. If we manage to form $\leq K$ segments this way, the threshold $\tau$ is feasible; otherwise, $\tau$ is set too low. Binary searching $\tau$ (which ranges from the cost of the largest single sequence up to the cost of all sequences) finds the best maximum cost for this contiguous partitioning problem in $\mathcal{O}(N \log \mathcal{C}(\mathcal{T}))$ time, which is very fast in practice. We note that computing the cost of a segment (group of sequences) can be done incrementally during the greedy scan: when adding a sequence in DFS order, only the suffix beyond its longest common prefix with the previous sequence contributes new tree tokens. Algorithm 4 summarizes this load-balanced partitioning procedure.

## 5. Evaluation

We conduct a comprehensive evaluation of AREAL-DTA to investigate the following two core research questions:

- *RQ1: Does* AREAL-DTA *introduce end-to-end RL training speedup when compared with the state-of-the-art asynchronous RL training system?*

- *RQ2: How effective is each key component in* AREAL-DTA *to improve the efficiency of the training GPUs?*

### 5.1. Experimental Setup

We select the state-of-the-art implementation AREAL (Fu et al., 2025) as the dense baseline and use $\tau^2$-bench as our RL training workflow. As shown in Figure 1, multi-turn trajectories in $\tau^2$-bench share the global system prompt and reuse previous prompts/responses across later turns. Under the definition $C = \eta_{\text{tokens}}/\eta_{\text{tree tokens}}$, the raw $\tau^2$-bench rollouts have a full-tree compression rate of $9.43\times$, corresponding to an $89.4\%$ prefix-sharing rate. If rollouts that are prefixes of other rollouts are collapsed and only leaf sequences are counted, the leaf-only compression rate is still $5.56\times$ ($82.0\%$ prefix sharing). We run GRPO (Shao et al., 2024) as our RL training algorithm. For all model scales, we follow the standard QWEN-3 architecture, set the model-training context length to 16K, and run experiments on an 8-GPU instance equipped with Nvidia H800 GPUs. We evaluate Qwen3-1.7B, 4B, 8B, and 14B for training-throughput and memory ablations; the end-to-end RL training experiments use Qwen3-1.7B and 8B.

AReal employs standard dense causal attention, in which shared tokens across trajectories are computed repeatedly in each generated rollout sequence. We enable activation recomputation for AReal when needed to avoid out-of-memory errors at the 16K context length, denoted by "Dense+CKPT" in our ablations. We also include a Sparse baseline implemented with FlexAttention (Dong et al., 2024), which evaluates the packed prefix tree with an explicit tree attention mask. Sparse uses a 24K micro-batch token budget and a FlexAttention block size of 128. For AReal-DTA, the default chunk block size in chunked backpropagation is 2048. In the ablation figures, "DFS" denotes the efficient DFS traversal order described in Algorithm 1, and "LB" denotes the large-block variant with a chunk block size of 4096. For backward-pass ablations, we measure the time and peak memory of the gradient-computation step, excluding optimizer state updates.

## 5.2. End-to-End RL Training Performance (RQ1)

To answer RQ1, we present the following experimental results about the end-to-end RL training performance. Note that since AReal-DTA increases the training throughput of the policy model, the optimal allocation of the rollout GPU workers and the training GPU workers will be different. We present the manually tuned optimal parallel RL training configurations in Table 1. We further provide a sweep over rollout/training GPU allocations in Appendix B.1. End-to-end speedup is computed from the measured pipeline throughput under the selected rollout/training allocation.

*Table 1.* Optimal asynchronous RL training configurations for AReal-DTA and AReal under GRPO when training 1.7B and 8B models. We denote the parallel strategy for rollout generation and model training using the format dp$x$-pp$y$-tp$z$, where $x$, $y$, and $z$ represent the data-parallel, pipeline-parallel, and tensor-parallel degrees. In rollout generation, a data-parallel degree of $x$ means that identical rollout workers are replicated $x$ times.

| Model | System | Parallel Strategy |
|-------|--------|-------------------|
| 1.7B | AReal-DTA | dp5-pp1-tp1 + dp3-pp1-tp1 |
|       | AReal     | dp2-pp1-tp1 + dp6-pp1-tp1 |
| 8B   | AReal-DTA | dp5-pp1-tp1 + dp3-pp1-tp1 |
|       | AReal     | dp2-pp1-tp1 + dp6-pp1-tp1 |

**RL reward curve.** Figure 3 presents the reward curves of AReal-DTA and AReal on the $\tau^2$-bench dataset for the 1.7B and 8B models, using the RL training step as the x-axis. Due to the inherent non-determinism of asynchronous RL training, for the same model size, the reward curves in Figure 3 are similar but not identical between AReal-DTA and the AReal baseline. These results indicate that AReal-DTA's design does not compromise training stability in asynchronous RL training.

**End-to-end RL training throughput.** In Figure 4, we

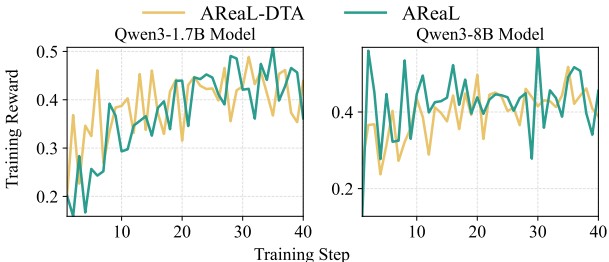

*Figure 3.* Across training steps, we present the reward comparison between AReal-DTA and AReal on the $\tau^2$-bench dataset and the 1.7B/8B models.

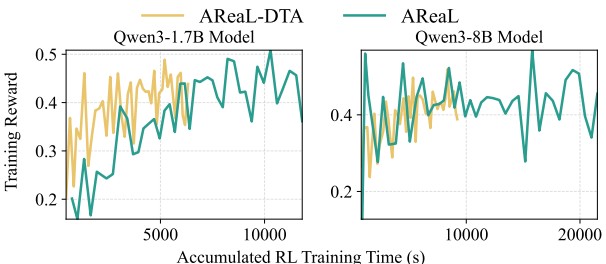

*Figure 4.* Across accumulated RL training time, we present the reward comparison between AReal-DTA and AReal on the $\tau^2$-bench dataset and the 1.7B/8B models.

present the reward curves of AReal-DTA and AReal using the accumulated real-world RL training time as the x-axis. Figure 4 demonstrates that AReal-DTA improves RL training efficiency: for the final RL training step in Figure 4, AReal-DTA achieves 1.28× and 2.28× end-to-end training throughput improvement compared to AReal on 1.7B and 8B models, respectively. As the training configurations summarized in Table 1 reveal, due to the superior model training performance, AReal-DTA can allocate more GPUs to the rollout generation phase, leading to improved end-to-end speedup. On the other hand, the lower training efficiency of the training GPU in AReal necessitates allocating more GPUs to model training, resulting in suboptimal overall performance.

## 5.3. Ablation Studies (RQ2)

**Ablation study on the backward pass optimizations.** We first isolate the single-GPU training efficiency of AReal-DTA by running all backward-pass variants on one GPU. We present ablation studies of AReal-DTA's backward pass optimizations in Figure 5 (throughput evaluations) and Figure 6 (Qwen3-1.7B GPU memory evaluations). Compared to Dense+CKPT, the Tree method avoids executing separate FWD/BWD passes for every rollout sequence and instead traverses only the unique segments in the prefix tree. This reduces both the number of effective FWD/BWD computations and the associated launch and graph-construction overhead relative to vanilla sequence-wise training. As a result, AReal-DTA achieves up to 7.53× speedup with

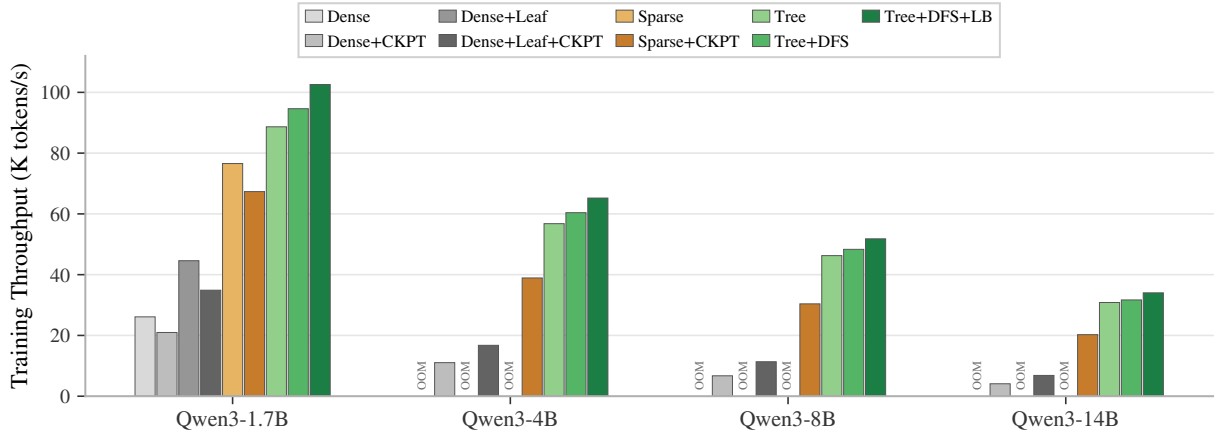

*Figure 5.* Single-GPU training-throughput ablation on $\tau^2$-bench. "Sparse" uses a FlexAttention tree mask (Dong et al., 2024); missing bars indicate OOM runs.

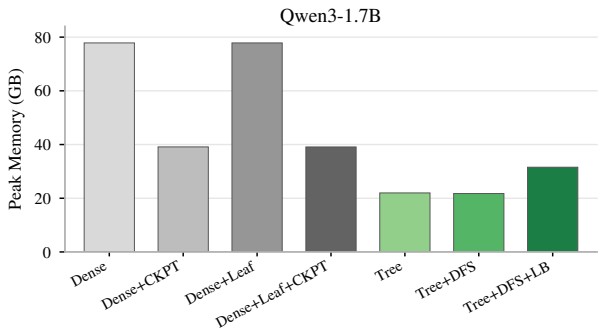

*Figure 6.* Single-GPU peak training memory ablation for Qwen3-1.7B on $\tau^2$-bench.

the Tree method; applying the optimized DFS traversal order (denoted by "DFS") further improves the speedup to $7.74\times$. With sufficient memory budget, using the larger chunk block size (denoted by "LB") further boosts the speedup to $8.31\times$. On Qwen3-1.7B, where Dense can run without OOM, AREAL-DTA reduces peak memory by over $50\%$ compared with Dense. Note that simply merging sequences that are fully contained as prefixes by other sequences can yield a $1.67\times$ speedup.

We also compare AREAL-DTA against Sparse and Sparse+CKPT. Sparse computes shared-prefix trajectories with an explicit tree mask, and therefore serves as a practical proxy for sparse tree attention mask methods. In 4B/8B/14B parameter models, activations for a single long sequence are already substantial, so Sparse needs activation recomputation to fit sequence-level training states in memory. On top of this, the packed tree tokens and their training states can still exceed GPU memory even with activation recomputation, so Sparse must split large prefix trees into smaller subtrees. Consequently, its effective compression rate drops to $7.47\times$ on $\tau^2$-bench, while AREAL-DTA operates on

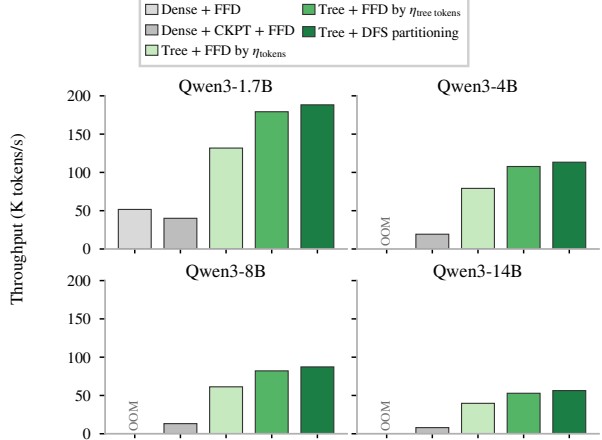

*Figure 7.* Backward throughput ablation with 2 GPUs.

the full prefix tree with $C = 9.43\times$. Sparse also pays the cost of executing an irregular tree mask, whose MFU is typically lower than optimized dense-attention kernels. In contrast, AREAL-DTA processes one active path at a time with highly optimized dense-attention kernels (i.e., a causal mask) and constructs chunk-level computation graphs from cached prefix KV states. As a result, AREAL-DTA achieves $1.52\times$–$1.70\times$ higher throughput over Sparse+CKPT across model sizes. We report additional low-compression workloads in Appendix B.2.

**Ablation study on the data-parallel balancing algorithm.** Figures 7, 8, and 9 compare our load-balanced partitioning against greedy data-parallel baselines that assign training data to the worker with the smallest current workload, measured either by total sequence tokens or by $\mathcal{C}(\mathcal{T})$. Our method instead sorts sequences by the DFS order of the prefix tree $\mathcal{T}$ and performs contiguous partitioning evenly based on $\mathcal{C}(\mathcal{T})$. This reduces extra tree tokens introduced by

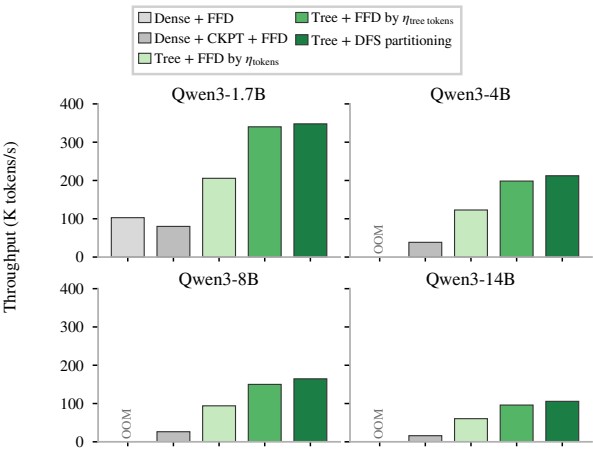

*Figure 8.* Backward throughput ablation with 4 GPUs.

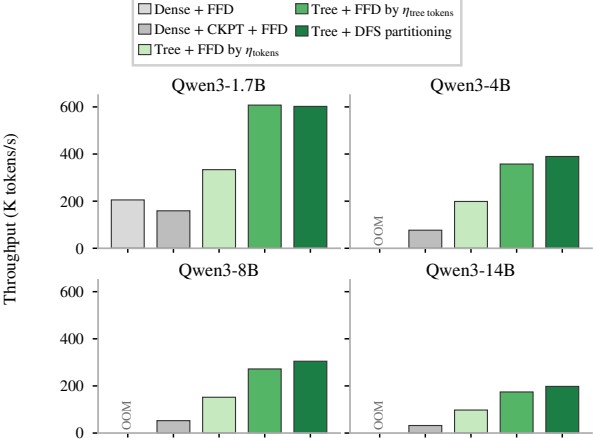

*Figure 9.* Backward throughput ablation with 8 GPUs.

partitioning and lowers wall-clock time; disabling the load-balanced partitioning degrades performance by 11.93%.

### 5.4. Additional Discussion

We conduct additional experiments to further characterize the efficiency and applicability of AReaL-DTA. The details are included in Appendix B. Key findings are summarized below: First, the rollout/training GPU allocation sweep further shows that *improving policy-model training throughput changes the optimal asynchronous pipeline balance*: AReaL-DTA achieves its best end-to-end throughput with a 5/3 rollout/training GPU split for both 1.7B and 8B models, whereas AReaL peaks at 2/6. Second, the low-compression experiments clarify the boundary of AReaL-DTA: *the benefits of* AReaL-DTA *are most significant when rollouts exhibit substantial prefix sharing*, while workloads with compression rates close to one provide limited reuse opportunities and may expose traversal overhead, especially for smaller models. These results support the main

conclusion that AReaL-DTA is particularly effective for prefix-sharing-heavy RL workloads, while its advantage depends on the available prefix reuse in the training data.

## 6. Conclusion

The efficiency of RL post-training can be improved by reducing redundant computation because many workflows generate rollout trajectories that share long prefixes. In this paper, we presented AReaL-DTA, a system that exploits prefix sharing to improve the efficiency and scalability of RL training. AReaL-DTA organizes rollouts as a prefix tree and performs a dynamic DFS traversal that reuses shared-prefix computation while interleaving forward and backward passes, keeping the live computation state bounded by the longest root-to-leaf path rather than the total tree size. To scale out to multi-GPU training in asynchronous RL frameworks, AReaL-DTA further introduces a load-balanced dispatch strategy that batches rollouts into multiple prefix trees and distributes them across trainer GPUs to minimize GPU idle time while preserving prefix reuse. In our experiments on $\tau^2$-bench, AReaL-DTA improves training throughput by up to $8.31\times$ and end-to-end RL throughput by up to $2.28\times$, enabling larger group sizes per prompt and more rollouts per iteration under the same hardware budget.

For future work, we expect several interesting directions that can further improve AReaL-DTA. First, adaptive run-time policies could estimate prefix-sharing and attention-compression rates online to decide when dynamic tree execution should be used, thereby reducing overhead on low-sharing workloads. Second, more general scheduling strategies could jointly optimize rollout generation, prefix-tree construction, and trainer allocation as workload characteristics evolve during asynchronous RL training. Lastly, extending AReaL-DTA to broader RL algorithms, longer-horizon agentic tasks, and larger model scales would further clarify its applicability and guide future system-level optimizations for efficient LLM post-training.

## Acknowledgements

This work is supported by the Ant Group-Tsinghua University joint project, Ant Group-HKUST joint project, the HKUST startup grant R9895 from CSE; RGC-ECS project 26218024; RGC-NSFC project CRS HKUST601/24.

## Impact Statement

This paper presents work whose goal is to advance the field of Machine Learning. There are many potential societal consequences of our work, none of which we feel must be specifically highlighted here.

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

# A. Dynamic Tree Attention Pseudocode

Algorithms 1–4 give the pseudocode for AREAL-DTA. The notation and primitives used by the algorithms are summarized below.

## Notation and primitives

1: Assume all rollout sequences are located at leaf nodes of the prefix tree $\mathcal{T}$.
2: $\mathrm{FWD}(\ell_{\mathrm{start}} \rightarrow \ell_{\mathrm{end}})$: run a regular forward pass over the token segment $[\ell_{\mathrm{start}} + 1, \ell_{\mathrm{end}}]$ and construct the computation graph.
3: $\mathrm{FWD\text{-}NOGRAD}(\ell_{\mathrm{start}} \rightarrow \ell_{\mathrm{end}})$: run a forward pass only to materialize detached KV-cache state.
4: $\mathrm{BWD}(\ell_{\mathrm{backward}} \rightarrow \ell_{\mathrm{lcp}})$: backpropagate attached losses and incoming gradients from $\ell_{\mathrm{backward}}$ to $\ell_{\mathrm{lcp}}$ using the constructed graph.
5: $\mathrm{LCP}(\mathbf{s}, \mathbf{s}') \leftarrow \max\{\ell : \mathbf{s}_{1:\ell} = \mathbf{s}'_{1:\ell}\}$.
6: Active stack $S$: stores the currently materialized root-to-leaf token/KV-cache path.

---

**Algorithm 1** Backward DFS Order

---

**Require:** Prefix tree $\mathcal{T}$
 1: **for** node $v$ in postorder$(\mathcal{T})$ **do**
 2:   **if** $v$ is a leaf **then**
 3:     $\mathrm{tail}(v) \leftarrow \mathrm{depth}(v)$
 4:   **else**
 5:     order children $u$ increasingly by $\big(\mathbf{1}[u \text{ is internal}], \mathrm{tail}(u)\big)$
 6:     *leaf child first; among the same type, smaller tail first*
 7:     *with an infinite block size, the reversed DFS order places short/leaf-only branches late*
 8:     *so they can be popped directly without extra* FWD-NOGRAD *cache anchors*
 9:     $\mathrm{tail}(v) \leftarrow \mathrm{tail}(\text{first child under this order})$
10:   **end if**
11: **end for**
12: $\rho \leftarrow$ leaf order produced by DFS using the child order above
13: **return** reverse$(\rho)$

---

**Algorithm 2** Vanilla Dynamic Tree Attention Backward

---

**Require:** Prefix tree $\mathcal{T}$, losses $\{\mathcal{L}(u)\}$ for leaf nodes
 1: initialize active stack $S$
 2: **procedure** DFS$(u)$
 3: **if** $u$ is a leaf **then**
 4:   attach $\mathcal{L}(u)$ at $\mathrm{depth}(u)$
 5: **end if**
 6: **for** child $v$ of $u$ **do**
 7:   $\mathrm{FWD}(\mathrm{depth}(u) \rightarrow \mathrm{depth}(v))$
 8:   push token segment $[\mathrm{depth}(u) + 1, \mathrm{depth}(v)]$ onto $S$
 9:   DFS$(v)$
10:   $\mathrm{BWD}(\mathrm{depth}(v) \rightarrow \mathrm{depth}(u))$          *propagate attached losses and incoming gradients*
11:   pop token segment $[\mathrm{depth}(u) + 1, \mathrm{depth}(v)]$ from $S$
12: **end for**
13: **end procedure**
14: DFS$(\mathrm{root}(\mathcal{T}))$

---

---

**Algorithm 3** Dynamic Tree Attention Backward

---

**Require:** Prefix tree $\mathcal{T}$, losses $\{\mathcal{L}(\mathbf{s})\}$, block size $B$
1: **procedure** POP_BYCHUNK($\ell_{\text{backward}} \to \ell_{\text{lcp}}$)
2: choose evenly spaced boundaries $t_0 = \ell_{\text{lcp}} < t_1 < \cdots < t_m = \ell_{\text{backward}}$ such that $t_{j+1} - t_j \leq B$
3: **for** $k = m - 1, \ldots, 0$ **do**
4:     FWD($t_k \to t_{k+1}$)                              *construct the computation graph for this segment*
5:     BWD($t_{k+1} \to t_k$)                           *propagate incoming gradients and attached losses in the segment*
6:     pop suffix within the range $[t_k + 1, t_{k+1}]$       *remove the popped tokens*
7: **end for**
8: **end procedure**

9: $\pi \leftarrow$ BACKWARDDFSORDER($\mathcal{T}$); initialize active stack $S$ and $\ell_{\text{backward}} \leftarrow 0$
10: **for** $i = 0, \ldots, |\pi| - 1$ **do**
11:     $\mathbf{s} \leftarrow \pi_i$; $n \leftarrow |\mathbf{s}|$
12:     **if** $i > 0$ **then**
13:         $\ell_{\text{lcp}} \leftarrow$ LCP($\pi_{i-1}, \mathbf{s}$)
14:         POP_BYCHUNK($\ell_{\text{backward}} \to \ell_{\text{lcp}}$)
15:     **else**
16:         $\ell_{\text{lcp}} \leftarrow 0$
17:     **end if**
18:     FWD($\ell_{\text{lcp}} \to n$); push suffix $\mathbf{s}[\ell_{\text{lcp}} + 1 : n]$ onto $S$
19:     attach $\mathcal{L}(\mathbf{s})$ at position $n$; $\ell_{\text{backward}} \leftarrow n$
20:     **if** $i + 1 < |\pi|$ **then**
21:         $\ell_{\text{next\_anchor}} \leftarrow$ LCP($\mathbf{s}, \pi_{i+1}$)
22:     **else**
23:         $\ell_{\text{next\_anchor}} \leftarrow 0$
24:     **end if**
25:     $p_{\text{pop}} \leftarrow n - \ell_{\text{next\_anchor}}$
26:     **if** $p_{\text{pop}} \geq B$ **then**
27:         $m \leftarrow \lceil p_{\text{pop}}/B \rceil$; $\Delta \leftarrow \lceil p_{\text{pop}}/m \rceil$
28:         $\ell_{\text{next\_anchor}} \leftarrow n - \Delta$
29:     **end if**
30:     **if** $\ell_{\text{lcp}} < \ell_{\text{next\_anchor}}$ **then**
31:         FWD-NOGRAD($\ell_{\text{lcp}} \to \ell_{\text{next\_anchor}}$)       *materialize cache anchor for the next*
                                                                     *POP_BYCHUNK's computation graph construction*
32:     **end if**
33: **end for**
34: POP_BYCHUNK($\ell_{\text{backward}} \to 0$)

---

---

**Algorithm 4** Load-Balanced DFS-Contiguous Partitioning

---

**Require:** Rollout sequences $\mathcal{S}$, number of trainer GPUs $K \leq |\mathcal{S}|$
**Ensure:** Contiguous rollout groups $\mathcal{G}_1, \ldots, \mathcal{G}_K$
1: $\pi \leftarrow$ lexicographic order of $\mathcal{S}$                                            *a DFS leaf order of the prefix tree*
2: $N \leftarrow |\pi|$
3: **procedure** FEASIBLE$(\tau)$
4: **if** $\max_i |\pi_i| > \tau$ **then**
5:     **return** (FALSE, $\emptyset$)
6: **end if**
7: $m \leftarrow 1; a \leftarrow 1; c \leftarrow |\pi_1|;$ cuts $\leftarrow \emptyset$
8: **for** $i = 2, \ldots, N$ **do**
9:     $\delta \leftarrow |\pi_i| - \text{LCP}(\pi_{i-1}, \pi_i)$                                  *new tree tokens added by $\pi_i$*
10:     **if** $c + \delta > \tau$ **then**
11:         append cut $(a, i - 1); m \leftarrow m + 1; a \leftarrow i; c \leftarrow |\pi_i|$
12:     **else**
13:         $c \leftarrow c + \delta$
14:     **end if**
15: **end for**
16: append cut $(a, N)$
17: **return** $(m \leq K, \text{cuts})$
18: **end procedure**

19: $L \leftarrow \max_i |\pi_i|; R \leftarrow |\pi_1| + \sum_{i=2}^{N} \left( |\pi_i| - \text{LCP}(\pi_{i-1}, \pi_i) \right)$
20: **while** $L < R$ **do**
21:     $\tau \leftarrow \lfloor (L + R)/2 \rfloor$
22:     $(ok, \_) \leftarrow$ FEASIBLE$(\tau)$
23:     **if** $ok$ **then**
24:         $R \leftarrow \tau$
25:     **else**
26:         $L \leftarrow \tau + 1$
27:     **end if**
28: **end while**
29: $(\_, \text{cuts}) \leftarrow$ FEASIBLE$(L)$
30: **while** $|\text{cuts}| < K$ **do**
31:     split any non-singleton interval in cuts into two contiguous intervals
32: **end while**
33: **return** contiguous groups induced by cuts

---

*Table 2.* End-to-end throughput sweep under different rollout/training GPU allocations. A split of $X/Y$ denotes $N_{\text{roll}} = X$ rollout GPUs and $N_{\text{train}} = Y$ training GPUs. Throughput is reported in K tokens/s, and the best split for each model/system pair is bolded.

| Model | System | 1/7 | 2/6 | 3/5 | 4/4 | 5/3 | 6/2 | 7/1 |
|-------|--------|-----|-----|-----|-----|-----|-----|-----|
| Qwen3-1.7B | AREAL | 40.8 | **52.1** | 43.8 | 42.2 | 34.9 | 26.1 | 13.5 |
| Qwen3-1.7B | AREAL-DTA | 41.1 | 56.8 | 58.3 | 62.0 | **66.7** | 49.4 | 40.3 |
| Qwen3-8B | AREAL | 14.5 | **22.6** | 20.5 | 17.2 | 13.8 | 9.0 | 5.1 |
| Qwen3-8B | AREAL-DTA | 10.9 | 27.5 | 36.3 | 37.0 | **51.4** | 47.0 | 21.5 |

## B. Additional Experimental Results

### B.1. Rollout/Training GPU Allocation Sweep

The end-to-end throughput of an asynchronous RL pipeline depends on the balance between rollout generation and policy-model training. Let the total number of GPUs be $N$, and let $N_{\text{roll}}$ and $N_{\text{train}} = N - N_{\text{roll}}$ denote the number of rollout and training GPUs, respectively. If the ideal full-cluster throughputs of the training and rollout stages are $T_{\text{train}}$ and $T_{\text{roll}}$, an idealized pipeline throughput can be written as

$$T_{\text{e2e}}(N_{\text{roll}}) = \min\left(T_{\text{train}}\frac{N_{\text{train}}}{N}, T_{\text{roll}}\frac{N_{\text{roll}}}{N}\right). \quad (3)$$

This formulation explains why improving policy-model training throughput can shift the optimal configuration toward allocating more GPUs to rollout generation. Table 2 reports a measured sweep under the same 8-GPU setting as the main experiments. The best allocation for AREAL-DTA is $N_{\text{roll}}/N_{\text{train}} = 5/3$ for both 1.7B and 8B models, while AREAL peaks at 2/6. This directly supports the configurations used in Table 1: prefix reuse is already exploited by the rollout engine, whereas AREAL-DTA introduces the corresponding sharing on the training side and changes the pipeline balance.

### B.2. Throughput Under Low Compression Rates

The main experiments focus on $\tau^2$-bench, whose rollout trajectories contain a substantial amount of shared prefixes. To evaluate the boundary of AREAL-DTA under weaker prefix sharing, we additionally measure training throughput on three low-compression settings in Table 3: Modified Prefix, GSM8K (Cobbe et al., 2021), and AIME (Mathematical Association of America, 2026). We report both the compression rate $C = \eta_{\text{tokens}}/\eta_{\text{tree tokens}}$, which captures the amount of unique-token reuse in the prefix tree, and the attention compression rate $C_{\text{attn}}$, which measures the effective reuse in attention computation. Since $C_{\text{attn}}$ is usually lower than $C$, low-compression workloads reduce the benefit of tree execution, especially for smaller models where kernel-launch overhead and memory-bound effects are more visible.

These results show that AREAL-DTA is most effective when the workload contains meaningful prefix sharing. When $C$ is close to 1, shared-prefix reuse provides limited savings, while the dynamic traversal still pays overhead from extra graph-construction FWD passes. In these low-compression settings, AREAL-DTA can be slower than dense training for smaller models, where the traversal overhead is more visible.

*Table 3.* Training throughput on low-compression workloads. Throughput is reported in K tokens/s on one GPU, with relative throughput over AREAL in parentheses. "Modified Prefix" removes the common global prompt from $\tau^2$-bench rollouts and keeps the leaf sequences. Activation recomputation is enabled for Qwen3-14B in Modified Prefix and GSM8K, and for all models in AIME.

| Setting | System | Qwen3-1.7B | Qwen3-4B | Qwen3-8B | Qwen3-14B | Compression |
|---|---|---|---|---|---|---|
| Modified Prefix $L_{\mathrm{avg}} \approx 1.6\mathrm{K}$ | AREAL | 28.16 (1.00×) | 14.19 (1.00×) | 9.38 (1.00×) | 4.35 (1.00×) | $C = 1.42, C_{\mathrm{attn}} = 1.17$ |
| | AREAL-DTA | 16.31 (0.58×) | 10.96 (0.77×) | 8.82 (0.94×) | 5.69 (1.31×) | |
| GSM8K $L_{\mathrm{avg}} \approx 1.8\mathrm{K}$ | AREAL | 27.57 (1.00×) | 13.88 (1.00×) | 9.16 (1.00×) | 4.33 (1.00×) | $C = 1.09, C_{\mathrm{attn}} = 1.01$ |
| | AREAL-DTA | 17.10 (0.62×) | 10.60 (0.76×) | 7.89 (0.86×) | 4.72 (1.09×) | |
| AIME $L_{\mathrm{avg}} \approx 16\mathrm{K}$ | AREAL | 16.50 (1.00×) | 7.85 (1.00×) | 5.72 (1.00×) | 3.56 (1.00×) | $C = 1.008, C_{\mathrm{attn}} = 1.00$ |
| | AREAL-DTA | 19.60 (1.19×) | 8.72 (1.11×) | 5.96 (1.04×) | 3.39 (0.95×) | |

