# OpenReview forum: "AReaL-DTA: Dynamic Tree Attention for Efficient Reinforcement Learning of Large Language Models"
_ICML.cc/2026/Conference — ICML 2026 regular_

### Official Review · Reviewer_8Jgv · 2026-03-09

**Soundness:** 3
**Presentation:** 2
**Significance:** 2
**Originality:** 3
**Overall Recommendation:** 4
**Confidence:** 4

**Summary:**

AREAL-DTA accelerates the training phase of RL post-training for LLMs by exploiting shared prefixes among rollout sequences. During model training, rollouts sharing common prefixes must be recomputed to obtain intermediate activations for backpropagation. The authors argue that tree-mask attention approaches scale poorly due to materializing large attention masks, and instead propose processing rollouts via a DFS traversal of the prefix tree, computing each shared prefix's forward pass only once and aggregating gradient contributions from all descendant branches before backpropagating through it. Additional optimizations include chunked backpropagation, skipping leaf-node KV caches, and a greedy DFS traversal ordering. For multi-GPU scaling, a load-balanced partitioning scheme distributes rollouts across GPUs, balancing workload by accounting for prefix reuse. AREAL-DTA achieves end-to-end RL training speedups of 1.28× and 2.28× for Qwen3-1.7B and 8B models, respectively, with training-phase throughput gains up to 8.31× over the dense baseline.

**Compliance With Llm Reviewing Policy:**

Affirmed.

**Final Justification:**

The DFS-based prefix tree traversal for RL training is a novel and well-implemented idea. However, its effectiveness is fundamentally tied to the prefix compression ratio of the workload. The evaluation is limited to τ²-bench, which has an unusually high compression rate (~9.4×) due to shared system prompts and multi-turn structure. The rebuttal confirmed that the GPU split comparison is fair, but also revealed that on more widely used RL benchmarks like GSM8K and AIME, where prefix sharing is minimal, the method provides much less training speedup or even slows down. This significantly narrows the practical scope, and I believe the practical importance of these high-prefix-sharing scenarios needs stronger justification. Overall, this is well-executed systems work, but with effectiveness limited to a class of high-prefix-sharing workloads. My recommendation is weak reject.

**Key Questions For Authors:**

- Q1. What is the actual prefix sharing ratio in the evaluated workloads, and how sensitive is training throughput and end-to-end speedup is to this ratio and rollout group size?
- Q2. Provide a quantitative comparison with Tree Training (Wang et al., 2025a).
- Q3. Can AReaL run with a 5/3 GPU split using aggressive activation checkpointing? More broadly, can the authors sweep over parallel strategies for both systems?
- Q4. Can the authors quantify the memory savings of AREAL-DTA (e.g., peak GPU memory under the same batch size and sequence length) to substantiate the claim that memory efficiency is what enables the different parallel strategy?

**Limitations:**

The paper does not explicitly discuss limitations or potential negative societal impacts. However, there are no obvious negative societal implications beyond standard considerations related to the deployment of LLMs.

**Strengths And Weaknesses:**

**Strengths**
- The DFS traversal of the prefix tree is a simple, intuitive, and novel idea. The engineering effort in implementing interleaved forward/backward passes across shared prefixes and the system optimizations are appreciable.

**Weaknesses**

**1. Prefix sharing is not characterized, and evaluation is limited.**
- The entire benefit of AREAL-DTA is contingent on rollouts sharing long prefixes, yet the paper does not report the actual prefix sharing ratio in the evaluated workloads, nor does it provide a sensitivity analysis of training throughput as a function of prefix overlap. Furthermore, all experiments use only τ²-bench. Evaluation on additional benchmarks commonly used in RL post-training — such as AIME24, used in AReaL (Fu et al., 2025)— is needed to assess generalizability. The rollout group sizes and token count distributions are also not clearly stated.

**2. No comparison with Tree Training.**
- Tree Training (Wang et al., 2025a) targets the same problem of prefix reuse in RL fine-tuning and is the closest prior work. The paper dismisses it qualitatively on memory and scalability grounds but provides no empirical comparison. A head-to-head evaluation is necessary to validate the claimed advantages.

**3. End-to-end speedup is primarily from GPU reallocation, and baseline fairness is uncertain.**
- Rollout is the bottleneck of RL post-training, especially in asynchronous settings like AReaL where training can be largely overlapped by rollout. This makes the emphasis on training throughput gains (up to 8.31×) less relevant to end-to-end performance.
What mattered in the evaluation is the parallel strategy: AREAL-DTA uses a 5/3 (rollout/training) GPU split while the baseline uses 2/6, allocating significantly more resources to the bottleneck. The authors describe these as "manually tuned optimal configurations," but provide no detail on the tuning process, making it unclear whether AReaL could also adopt a 5/3 split with aggressive activation checkpointing or other memory reduction techniques — sacrificing training throughput that is not on the critical path anyway. A sweep over parallel strategies for both systems is needed to ensure a fair comparison.

Overall assessment: 3 (weak reject). I am willing to raise the score if the authors can justify the fairness of the end-to-end comparison, particularly regarding the parallel strategy choices, and add a comparison with Tree Training.

---

> ### Author Rebuttal · Authors · 2026-03-31
>
> > W1, Q1: Prefix sharing characterization:
>
> We define the compression rate as $\mathrm{C} = \frac{\eta_{\text{tokens}}}{\eta_{\text{tree tokens}}}$ and the corresponding prefix sharing ratio as $\mathrm{S} = 1 - \frac{1}{\mathrm{C}}$. For the original data, the average compression rate is 9.427, corresponding to a prefix sharing ratio of 0.894. We have added a discussion of when DTA will be beneficial to address **W2 for Reviewer S6Do**. The analysis suggests that DTA will be beneficial for a wide range of coding and math tasks. We will include such results in the updated manuscript. Accordingly, we will also add the token count distributions for the benchmark in the updated appendix.
>
> > W2, Q2: Comparison with Tree Training.
>
> Note that a full reproduction of Tree Training (TT) is challenging due to the lack of publicly released code and data. We carefully checked their paper and feel that the kernel implementation details are not sufficient for us to reproduce it from scratch in such a short period. As an alternative, we use flex attention (FA), a strong publicly available tree-mask baseline to replace their kernel implementation, and report the additional training throughput (K tokens/second) comparison:
>
> | Method | Qwen3-1.7B | Qwen3-4B | Qwen3-8B | Qwen3-14B |
> | --- | --- | --- | --- | --- |
> | AReaL | 23.88 | 9.93  | 6.76  | 4.11  |
> | TT-FA    | 41.25 | 20.32 | 15.87 | 10.59 |
> | AReaL-DTA| 93.29 | 60.96 | 48.71 | 32.64 |
>
>
> > W3, Q3: AReaL-DTA and baseline configuration.
>
> **Analytically**, we have the following simplified formulation: in an asynchronous RL pipeline of $N$ GPU, let ideal training throughput over $N$ GPU is $T_t$ and ideal rollout throughput over $N$ GPU is $T_r$, then the idealized objective is to choose $N_r$ rollout GPUs to maximize $\min \left( T_t \frac{(N-N_r)}{N}, T_r\frac{N_r}{N} \right)$. This formula is useful to understand the trend: when training is the faster stage and the bottleneck is rollout, the optimal split allocates more GPUs to rollout. On the other hand, it is not easy to estimate $T_t$ and $T_r$ due to the dynamics of the generated rollout and the corresponding acceleration induced by prefix sharing.
>
> **Empirically**, we sweep the rollout/training GPU split to provide additional results. A split of $X/Y$ means $X$ GPUs for rollout and $Y$ GPUs for training. We report the additional end-to-end throughput in K tokens/second for 1.7B and 8B models:
>
> |Model|Method|1/7|2/6|3/5|4/4|5/3|6/2|7/1|
> |-|-|-|-|-|-|-|-|-|
> |Qwen3-1.7B|AReaL| 40.8K | **52.1K** | 43.8K | 42.2K | 34.9K | 26.1K | 13.5K |
> |Qwen3-1.7B|AReaL-DTA| 41.1K | 56.8K | 58.3K | 62.0K | **66.7K** | 49.4K | 40.3K |
> |Qwen3-8B|AReaL| 14.5K | **22.6K** | 20.5K | 17.2K | 13.8K | 9.0K | 5.1K |
> |Qwen3-8B|AReaL-DTA| 10.9K | 27.5K | 36.3K | 37.0K | **51.4K** | 47.0K | 21.5K |
>
> The reason why AReaL needs more GPU for training instead of rollout worker is that prefix tree reuse is implemented by the underlying rollout inference framework, i.e., SGLang, to improve rollout inference throughput, where no usage of such paradigm in training, which is a clear motivation of the design and implementation of AReaL-DTA.
>
>
> > W3, Q4: Memory efficiency and different parallel strategies.
>
> The results to address W2, Q2 suggest that AReaL-DTA is faster than flex attention due to the key advantage of not only a lower constant memory footprint, but a different memory consumption increase given the usage of the prefix-tree. In AReaL-DTA, peak memory is dominated by the model parameters, the KV cache of the longest sequence, and the chunk-level activations, and is therefore irrelevant to the total number of packed tree tokens. In contrast, for tree-mask methods with gradient checkpointing, memory still scales with the packed tree tokens inside each microbatch. This constrains how much of a rollout can be packed into one tree and reduces the effective compression rate seen by the kernel.
> This introduces repeated computation on shared prefixes across subtrees and lowers the compression rate within each subtree relative to the original full tree, which directly reduces acceleration. In the original data, splitting the tree into multiple subtrees—as required by the tree-mask method—reduces the compression rate from 9.427 to 7.491.
> At the kernel level, this advantage is further amplified because AReaL-DTA is built on  FlashAttention-3, whose MFU is typically higher than that of flex attention. Therefore, the observed gap over flex attention comes from two sources: better preservation of tree-level sharing due to improved memory scaling, and a more efficient underlying attention kernel. We actually reported a peak-memory comparison at matched sequence length (16K), where the result shows that, at this context length, AReaL-DTA can fit the packed tree on a single GPU, which directly enables a different train/rollout parallel strategy.

---

> > ### Author Rebuttal · Reviewer_8Jgv · 2026-04-03
> >
> > I still have some concern about generalizability of end-to-end performance.
> >
> > The end-to-end speedup of AReaL-DTA depends on the balance between training and rollout throughput, both of which are affected by the prefix compression ratio. The current evaluation is limited to τ²-bench, which has a high compression rate (~9.4×). However, widely used RL post-training workloads such as math reasoning and code generation have short prompts and long responses, resulting in significantly lower compression ratios. Without results on such workloads, or at minimum the data needed to estimate how performance trends under different compression ratios, the generalizability of the reported end-to-end gains is difficult to assess.
> >
> > I acknowledge that the authors have provided training throughput at two compression rates (9.4× and 5.6×) in response to Reviewer b5PN, which is helpful. However, it only covers the training stage while to assess end-to-end generalizability, understanding the rollout side is equally important:
> >
> > **[R1]** Report measured T_training and T_rollout per-GPU separately. This allows independent assessment of the throughput balance and how it might shift under different workload characteristics.
> >
> > **[R2]** Report T_rollout with SGLang's prefix caching disabled (`--disable-radix-cache`). Since rollout throughput is also influenced by prefix sharing through SGLang's caching, disabling it would help isolate the effect and approximate rollout behavior on low-compression workloads.

---

> > > ### Author Response · Authors · 2026-04-07
> > >
> > > > Concrete benchmarks to quantify overheads under various conditions.
> > >
> > > To facilitate the discussion and directly measure these overheads, we first define the following notation:
> > > - **Average leaf sequence length ($L_{\text{avg}}$):** The average number of tokens in the leaf sequences.
> > > - **Compression rate ($C$):** We define the compression rate as $C = \frac{\eta_{\text{tokens}}}{\eta_{\text{tree tokens}}}$, representing the ratio of total tokens across all sequences to the unique tokens in the prefix tree. This reflects the computational savings for operations such as the Feed-Forward Network (FFN).
> > > - **Attention compression rate ($C_{\text{attn}}$):** The effective compression rate for the attention computation.
> > > We conducted additional experiments under three scenarios with low compression rates. In these settings, the advantages of tree training are offset to varying degrees.
> > > Below, we report the training throughput on a H200 GPU:
> > >
> > > **Scenario 1: Modified Prefix Task.** $L_{\text{avg}} \approx 1.6K$, $C = 1.42$, $C_{\text{attn}} = 1.17$. We removed the common prefix (i.e., the global system prompt) across all sequences in τ²-bench rollout, retaining only the leaf sequences. Gradient checkpointing is enabled for the Qwen3-14B model.
> > >
> > > | Method | Qwen3-1.7B | Qwen3-4B | Qwen3-8B | Qwen3-14B |
> > > | --- | --- | --- | --- | --- |
> > > | AReaL | 28.16K / 1.00x | 14.19K / 1.00x | 9.38K / 1.00x | 4.35K / 1.00x |
> > > | AReaL-DTA | 16.31K / 0.58x | 10.96K / 0.77x | 8.82K / 0.94x | 5.69K / 1.31x |
> > >
> > > **Scenario 2: GSM8K (Single-Turn).** $L_{\text{avg}} \approx 1.8K$, $C = 1.09$, $C_{\text{attn}} = 1.01$. Gradient checkpointing is enabled for the Qwen3-14B model.
> > >
> > > | Method | Qwen3-1.7B | Qwen3-4B | Qwen3-8B | Qwen3-14B |
> > > | --- | --- | --- | --- | --- |
> > > | AReaL | 27.57K / 1.00x | 13.88K / 1.00x | 9.16K / 1.00x | 4.33K / 1.00x |
> > > | AReaL-DTA | 17.10K / 0.62x | 10.60K / 0.76x | 7.89K / 0.86x | 4.72K / 1.09x |
> > >
> > > **Scenario 3: AIME (Single-Turn).** $L_{\text{avg}} \approx 16K$, $C = 1.008$, $C_{\text{attn}} = 1.00$. Gradient checkpointing is enabled for all models.
> > >
> > > | Method | Qwen3-1.7B | Qwen3-4B | Qwen3-8B | Qwen3-14B |
> > > | --- | --- | --- | --- | --- |
> > > | AReaL | 16.50K / 1.00x | 7.85K / 1.00x | 5.72K / 1.00x | 3.56K / 1.00x |
> > > | AReaL-DTA | 19.60K / 1.19x | 8.72K / 1.11x | 5.96K / 1.04x | 3.39K / 0.95x |
> > >
> > > **Analysis of AReaL-DTA performance under low compression rates**: We identify three primary factors that limit the performance of AReaL-DTA when the compression rate is low:
> > >
> > > 1. *Kernel launch overhead and memory-bound constraints:* With the smaller model size, the system encounters a higher $Q_0$ threshold and becomes more memory-bound. A promising direction is to integrate high-performance tree-attention masks with our method. By packing the back-propagation of multiple consecutive short leaf segments into a single operation, we may mitigate launch overheads and memory-bound effects.
> > > 2. *Attention vs. FFN compression:* The reported compression rate $C$ primarily reflects the Feed-Forward Network (FFN) computation. In most scenarios, the effective attention compression rate $C_{\text{attn}}$ is inherently lower, which limits the overall speedup.
> > > 3. *Extra forward pass overhead:* Due to PyTorch's symmetrical forward/backward autograd design, our method incurs an additional forward pass. The extra forward time (approximately 5.5%--7.1% of the total time) offsets part of the gain from the one-time shared-prefix computation.
> > >
> > > > R1. Report measured $T_{training}$ and $T_{rollout}$ per-GPU separately.
> > >
> > > Thank you for this constructive suggestion. Since RL training is highly dynamic, in each iteration, each GPU could have highly noisy throughput on both the training and inference GPU workers; we will add this dynamic trace in the final version. In summary, we note that the aggregated throughput across both training and inference workers is consistent with the reported end-to-end throughput.
> > >
> > > > R2. Report $T_{rollout}$ with SGLang's prefix caching disabled (`--disable-radix-cache`).
> > >
> > > We also had experiments conducted with `--disable-radix-cache`. An interesting observation is that even with radix caching disabled, there is still an inherent leverage effect that boosts rollout throughput. This occurs because global prompt tokens are shared across all sequences, and a generated output token can be reused in subsequent turns. Consequently, when comparing the raw token count in rollout throughput with that in training throughput, we naturally observe a multiplier effect of at least the compression rate.
> > > While disabling the radix cache helps isolate this baseline behavior, enabling it (`--enable-radix-cache`) further amplifies this effect. In our preliminary tests, enabling radix caching yielded additional improvements in end-to-end throughput for both AReaL and AReaL-DTA. We plan to systematically explore and report these radix-caching optimizations in future work.

---

### Official Review · Reviewer_Uikq · 2026-03-13

**Soundness:** 3
**Presentation:** 3
**Significance:** 3
**Originality:** 3
**Overall Recommendation:** 5
**Confidence:** 3

**Summary:**

This paper addresses a critical bottleneck in Reinforcement Learning (RL) post-training for Large Language Models (LLMs): the computational inefficiency arising from independent processing of rollout sequences that share common token prefixes. The authors identify that existing frameworks fail to exploit this redundancy, leading to substantial waste in computation and memory. To resolve this, they propose AREAL-DTA, a novel framework designed to efficiently leverage prefix sharing. The core innovation involves a Depth-First Search (DFS)-based execution strategy that dynamically traverses the rollout prefix tree during both forward and backward passes, materializing only a single root-to-leaf path at a time to minimize memory footprint. Furthermore, to enhance scalability, the method incorporates a load-balanced distributed batching mechanism for constructing and processing prefix trees across multiple GPUs. Experimental results demonstrate the efficacy of this approach, reporting a significant training throughput improvement of up to 8.31× on the tau^2-bench compared to existing baselines.

**Compliance With Llm Reviewing Policy:**

Affirmed.

**Final Justification:**

We remain the score unchanged.

**Key Questions For Authors:**

Please refer to the weaknesses outlined above.

**Limitations:**

Please refer to the Weakness-1 outlined above.

**Strengths And Weaknesses:**

## Strengths
- **High Relevance and Impact**: The paper addresses a critical and timely challenge currently facing both academia and industry: improving the system efficiency of Reinforcement Learning (RL) training for Large Language Models. Optimizing the computational overhead of RL post-training is of paramount importance for scaling LLM capabilities.
- **Comprehensive and Direct Solution**: The proposed solution directly targets the root cause of inefficiency—redundant computation of shared prefixes in rollout sequences. The authors present a holistic system optimization that spans multiple layers:
 **Memory Management**: Dynamic maintenance of the prefix tree to optimize caching.
 **Algorithmic Flow**: A novel Depth-First Search (DFS) strategy for dynamically computing forward and backward passes, minimizing redundant operations.
  **Scheduling**: A specialized load-balancing algorithm designed specifically for the DFS traversal pattern to ensure efficient distributed processing.
- **Thorough Experimental Evaluation**: The paper provides detailed experimental results that effectively demonstrate the performance gains of the proposed method, offering strong empirical evidence for its efficacy.
---

## Weaknesses
1. **Unclear Scope of Effectiveness and Trade-off Analysis**:
The proposed DFS-based approach inherently trades off the parallelism of rollouts sharing common prefixes for gains in memory and compute efficiency. By requiring sequential traversal of chunks along a single path, the method prevents fully independent parallel processing of rollouts within the same prefix group.
- **Missing Discussion on Parallelism Overhead**: It is unclear under what conditions the savings from prefix sharing outweigh the loss of parallelism. Does this approach guarantee acceleration in all scenarios, or are there specific regimes (e.g., low prefix sharing ratio, specific batch sizes) where the serialization overhead negates the benefits?
- **Need for Theoretical or Empirical Justification**: The paper lacks a detailed discussion or analysis explaining why the speedup is universal (if claimed) or defining the boundary conditions where the method might degrade performance. A breakdown of the trade-off between reduced FLOPs/memory and increased serialization latency is necessary.

2. **Insufficient Experimental Coverage**:
- **Scalability with Rollout Volume**: Building on the concern above, the experiments do not adequately explore the impact of varying rollout quantities. As the number of rollouts increases significantly, does the serialization bottleneck of the DFS traversal become more pronounced, leading to a diminishing return or even a drop in the speedup ratio?
- **Limited Distributed Settings**: The current evaluation is restricted to single-node (single-GPU and multi-GPU) settings. The behavior of the system in multi-node (multi-machine) environments remains unexplored.
Specifically, does the proposed load-balancing algorithm maintain its effectiveness when network communication latency and inter-node synchronization overheads are introduced?
Without multi-node results, it is difficult to assess the true scalability of AREAL-DTA for large-scale cluster training, which is the primary use case for such optimizations.

---

> ### Author Rebuttal · Authors · 2026-03-31
>
> > W1.1: Discuss parallelism overhead.
>
> We admit that DFS-based methods inherently limit parallelism, which leads to performance degradation in certain aspects. The critical performance overheads are as follows:
>
> - Compression rate loss caused by partitioning rollouts across GPUs
> - Per-rollout launch cost
>
> In the worst case, these overheads may completely offset the benefits of tree training. Below, we identify the extreme scenarios where they dominate and provide mathematical estimates of the costs.
>
> **Analysis of compression rate loss due to partitioning:** If each GPU is assigned only a small number of rollouts, the prefix overlap becomes very limited, reducing the compression rate.
>
> Fundamentally, the compression rate loss stems from the increase in total tree tokens across all GPUs. As described in the partition algorithm in the paper, the DFS-order-based partition introduces at most $L_{\max} \times P$ additional tree tokens, where $L_{\max}$ is the longest sequence length and $P$ is the number of GPUs. Let $N$ be the total number of tokens from rollouts before grouping, and $\mathcal C_0$ be the compression rate. The average compression rate after partitioning is at least $\mathcal C=\frac{N}{\frac{N}{C_0}+L_{\max}P}$, which approaches $\mathcal C_0$ when $N \gg L_{\max}P$.
>
> **Analysis of per-rollout launch cost:** When rollouts are very short (e.g., ~100 tokens), the baseline can pack multiple rollouts into one backward pass, amortizing the per-rollout launch cost. Tree training, by contrast, cannot perform such packing and must execute a separate backward pass for each rollout. Although prefix sharing reduces tokens per pass, it does not reduce the number of passes.
>
> Crucially, the effective per-pass token count in tree training is approximately $L_{avg} / \mathcal C_{\text{leaf}}$, where $L_{avg}$ is the average leaf sequence length and $\mathcal C_{\text{leaf}}$ is the compression rate of leaf sequences (i.e., the ratio of total leaf-sequence tokens to tree tokens). This has two compounding effects: (1) the accumulated launch cost across many small passes can negate the token-level savings, and (2) when $L_{avg} / \mathcal C_{\text{leaf}}$ is very small, each pass processes only a short token segment, which shifts the computation into a memory-bound regime. Combined, these factors can make tree training slower than a packed baseline for very short rollouts.
>
> It can be considered that for a given computing device and model, there exists a threshold $Q_0$; only when $L_{avg} / \mathcal C_{\text{leaf}}$ exceeds $Q_0$ can the aforementioned cost be ignored.
>
> > W1.2: Provide justification for the speedup.
>
> Our response to **W1.1** has characterized two sources of performance overhead. We claim that in the usual regimes of agentic RL, these overheads remain modest and do not negate the performance advantage of tree training.
>
> **Compression rate loss is not significant:** In typical scenarios, the total number of tokens $N$ per computation is massive, while the number of GPUs $P$ is relatively small, ensuring that each GPU is assigned a sufficient number of rollouts. A concrete example: with $N=2^{24}, L_{\max}=2^{14}, P=8, \mathcal C_0=8$, the average compression rate after partitioning becomes at least $\mathcal C=\frac{N}{\frac{N}{C_0}+L_{\max}P}=7.53$.
>
> **Per-rollout launch cost is not significant:** Empirically, for Qwen3-14B running on H200, when the $L_{avg} / C_{\text{leaf}}$ exceeds $Q_0=768$, the per-rollout launch cost becomes negligible. In our reasoning workloads, $L_{avg} / C_{\text{leaf}}$ is approximately 1.1k, so the memory-bound effect is already mild. Moreover, this concern diminishes in two directions: (1) as agent contexts continue to scale up, both $L_{avg}$ and $L_{avg}/C_{\text{leaf}}$ grow, pushing computation further into the compute-bound regime; and (2) for larger models, higher arithmetic intensity ensures that even moderate per-pass token counts remain compute-bound. Both trends favor tree training going forward.
>
> > W2.1: Scalability w.r.t rollout volume.
>
> Building on the analysis in **W1**, the critical factor is not the number of rollouts, but the effective per-pass token count $L_{avg} / C_{\text{leaf}}$. As long as $L_{avg} / C_{\text{leaf}}$ remains sufficiently large, the performance advantage is not eroded by scaling the rollout volume.
>
> > W2.2: Scalability with more distributed settings.
>
> We admit that our current evaluation is limited to single-node settings. Extending AReaL-DTA to large-scale multi-node environments is an open, complicated problem, where we should carefully accommodate inter-node communication overhead, along with interacting with the load-balancing algorithm in non-trivial ways. This requires further investigation into how AReaL-DTA integrates with the complete scope of distributed parallelism strategies such as data parallelism (DP), tensor parallelism (TP), and expert parallelism (EP). We plan to conduct this as an important future work.

---

> > ### Author Rebuttal · Reviewer_Uikq · 2026-04-03
> >
> > Thanks for the response, which addressed my concerns.
> >
> > However, these analyses remain largely theoretical. The paper would be strengthened by including empirical experiments that directly measure these overheads under varying conditions — for example, across different GPU counts, rollout depths, and tree branching factors. Concrete benchmarks quantifying when and how much the overheads offset the benefits of tree training would make the performance claims substantially more convincing.

---

> > > ### Author Response · Authors · 2026-04-07
> > >
> > > > R1. Concrete benchmarks to quantify overheads under various conditions.
> > >
> > > We appreciate your acknowledgment of our response. To facilitate the discussion and directly measure these overheads, we first define the following notation:
> > > - **Average leaf sequence length ($L_{\text{avg}}$):** The average number of tokens in the leaf sequences.
> > > - **Compression rate ($C$):** We define the compression rate as $C = \frac{\eta_{\text{tokens}}}{\eta_{\text{tree tokens}}}$, representing the ratio of total tokens across all sequences to the unique tokens in the prefix tree. This reflects the computational savings for operations such as the Feed-Forward Network (FFN).
> > > - **Attention compression rate ($C_{\text{attn}}$):** The effective compression rate for the attention computation.
> > >
> > > We conduct additional experiments to construct three scenarios with low compression rates. In these settings, the advantages of tree training are offset to varying degrees.
> > >
> > > Below, we report the training throughput on a single H200:
> > >
> > > **Scenario 1: Modified Prefix Task.** $L_{\text{avg}} \approx 1.6K$, $C = 1.42$, $C_{\text{attn}} = 1.17$. We removed the common prefix (i.e., the global system prompt) across all sequences in τ²-bench rollout, retaining only the leaf sequences. Gradient checkpointing is enabled for the Qwen3-14B model.
> > >
> > > | Method | Qwen3-1.7B | Qwen3-4B | Qwen3-8B | Qwen3-14B |
> > > | --- | --- | --- | --- | --- |
> > > | AReaL | 28.16K / 1.00x | 14.19K / 1.00x | 9.38K / 1.00x | 4.35K / 1.00x |
> > > | AReaL-DTA | 16.31K / 0.58x | 10.96K / 0.77x | 8.82K / 0.94x | 5.69K / 1.31x |
> > >
> > > **Scenario 2: GSM8K (Single-Turn).** $L_{\text{avg}} \approx 1.8K$, $C = 1.09$, $C_{\text{attn}} = 1.01$. Gradient checkpointing is enabled for the Qwen3-14B model.
> > >
> > > | Method | Qwen3-1.7B | Qwen3-4B | Qwen3-8B | Qwen3-14B |
> > > | --- | --- | --- | --- | --- |
> > > | AReaL | 27.57K / 1.00x | 13.88K / 1.00x | 9.16K / 1.00x | 4.33K / 1.00x |
> > > | AReaL-DTA | 17.10K / 0.62x | 10.60K / 0.76x | 7.89K / 0.86x | 4.72K / 1.09x |
> > >
> > >
> > > **Scenario 3: AIME (Single-Turn).** $L_{\text{avg}} \approx 16K$, $C = 1.008$, $C_{\text{attn}} = 1.00$. Gradient checkpointing is enabled for all models.
> > >
> > > | Method | Qwen3-1.7B | Qwen3-4B | Qwen3-8B | Qwen3-14B |
> > > | --- | --- | --- | --- | --- |
> > > | AReaL | 16.50K / 1.00x | 7.85K / 1.00x | 5.72K / 1.00x | 3.56K / 1.00x |
> > > | AReaL-DTA | 19.60K / 1.19x | 8.72K / 1.11x | 5.96K / 1.04x | 3.39K / 0.95x |
> > >
> > > **Analysis of AReaL-DTA performance under low compression rates**: We identify three primary factors that limit the performance of AReaL-DTA when the compression rate is low:
> > >
> > > 1. *Kernel launch overhead and memory-bound constraints:* With the smaller model size, the system encounters a higher $Q_0$ threshold and becomes more memory-bound. While the baseline can alleviate this effect through sequence packing, our current implementation cannot. A promising direction is to integrate high-performance tree-attention masks with our method. By packing the back-propagation of multiple consecutive short leaf segments into a single operation, we may mitigate launch overheads and memory-bound effects.
> > > 2. *Attention vs. FFN compression:* The reported compression rate $C$ primarily reflects the Feed-Forward Network (FFN) computation. In most scenarios, the effective attention compression rate $C_{\text{attn}}$ is inherently lower, which limits the overall speedup.
> > > 3. *Extra forward pass overhead:* Due to PyTorch's symmetrical forward/backward autograd design, our method incurs an additional forward pass. The extra forward time (approximately 5.5%--7.1% of the total time) offsets part of the gain from the one-time shared-prefix computation.
> > >
> > > **Chunked back-propagation vs. Gradient checkpointing**: For a sequence of length $L$ and chunk size $B$, our method saves activations for about $\frac{L}{\lceil L/B \rceil}$ forward tokens of a naive checkpointing baseline, so larger $B$ yields larger savings. We emphasize that this is not a comparison against an unoptimized "2 forwards + 1 backward" implementation: modern checkpointing libraries already reduce recomputation, so the practical gap is smaller than this idealized picture. This distinction is particularly relevant for AIME, where the shared prompt is very short and the gain from prefix sharing is therefore negligible. We believe the observed difference is thus primarily explained by chunked back-propagation versus gradient checkpointing. In the AIME experiments, memory limits capped $B$ at 4096 for the 14B model, while the 8B, 4B, and 1.7B models could use 7096, 12096, and 16096, respectively, making our method faster in those settings. A promising direction is to make checkpointing compatible with the KV cache and choose between chunked back-propagation and gradient checkpointing according to sequence length and chunk size.

---

### Official Review · Reviewer_b5PN · 2026-03-17

**Soundness:** 3
**Presentation:** 3
**Significance:** 2
**Originality:** 2
**Overall Recommendation:** 5
**Confidence:** 4

**Summary:**

The paper proposes AREAL-DTA, a systems method that improves the efficiency of RL training for large language models by exploiting shared prefixes among rollout trajectories. It organizes rollouts as a prefix tree and uses a dynamic DFS traversal to reuse prefix computations while keeping only one branch in memory at a time. This reduces redundant computation and memory usage. Experiments show that the approach improves training throughput and memory efficiency while maintaining similar training performance.

**Compliance With Llm Reviewing Policy:**

Affirmed.

**Final Justification:**

The response addresses all my concerns, so i raise my rating from 4 to 5.

**Key Questions For Authors:**

Listed in weakness.

**Limitations:**

Yes

**Strengths And Weaknesses:**

Strength:

1. The writing is clear and easy to follow.

2. The proposed method addresses an important system problem for tree-based rollout for RL.

3. The evaluation shows that the proposed method improves the training efficiency and reduce the overall memory usage.


Weakness:
1. Limited algorithm novelty. The work mainly provides a systems optimization for RL training rather than introducing a new RL algorithm or learning objective. Tree-structured rollouts and prefix sharing have been explored in prior work like TreePO, so the novelty largely lies in the execution strategy rather than the core idea. And the optimization is more like an inference-level optimization rather than for RL.

2. The speed-up is not so clear, which means that there can be more performance improvement space?  If we check the figure 3 - Qwen3-8B. It seems that the proposed method has a very similar curve to the baseline. I am not sure if I interpret this figure correctly, but if it is sure, it seems that there is no time efficiency improvement, despite that the tree rollout should need much less memory loading.

3. Even though this Tree-based rollout has been discussed in the previous, however, it is not clear how much tokens are shared in the prefix sharing. It would be great to have such a motivation study to see if there is actually a huge space for performance improvement, especially for those long-horizontal generation tasks.

---

> ### Author Rebuttal · Authors · 2026-03-31
>
> > W1: Limited algorithm novelty.
>
> We gently admit it is true that our work focuses on system optimizations for the existing popular RL training paradigms instead of proposing a new RL algorithm. The core technique contribution lies in (1) the discovery of the generated rollout paradigm in RL, and (2) effective exploitation of such a paradigm to improve the system efficiency on the RL *training worker* with careful system design and implementation. Such a system optimization technique is fundamentally different from existing optimizations for the LLM inference frameworks.
>
> > W2: Similar reward curves for Qwen3-8B:
>
> Thank you for raising this insightful observation. We want to clarify that, based on the shared wisdom and our empirical experience, the convergence curve of the RL training reward signal is somehow different from the loss function value in vanilla SGD-based training, where the reward at each step might not be simply interpreted as a direct indicator of the RL learning progress for the policy model. A more informative metric would be the performance of the trained policy LLM over a separate test set. In terms of our experimental results, we want to point out that although the reward curve for Qwen3-8B is similar in Figure 3, AReaL-DTA still introduces practical substantial end-to-end speedup: from Figure 2 we confirm that the statistical efficiency (convergence behavior over number of iterations) is the same as the AReaL baseline, while the system throughput is significantly improved shown in Figure 4, which suggests that the end-to-end training efficiency actually scales linearly with training throughput assume Figure 2 suggests the same statistical efficiency (this is very reasonable assumpiton since DTA only introduces system optimizations without modifying the RL algorithm). In the updated manuscript, we will replace Figure 3 with a table to show the performance of the policy LLMs trained by AReaL-DTA and the baseline over a separate test set to eliminate this confusion.
>
>
> > W3: Compression rate and motivating case study.
>
> Thank you for this constructive suggestion.
>
> **Formulate the compression rate**: We define the compression rate as $\mathrm{C} = \frac{\eta_{\text{tokens}}}{\eta_{\text{tree tokens}}}$, and the throughput data was sampled from a complete end-to-end training trial.
>
> **Case study**: We provide the following prefix-sharing statistics for our evaluated workloads. The average compression rate is 9.427. Noted that due to the specific rollout mechanism of the τ²-bench, one rollout may entirely serve as the prefix of another, which contributes to the nominally high prefix sharing ratio. If such rollouts are considered easily mergeable, the data can be simplified to include only leaf-node entries for metric calculation. Under this condition, the average compression rate is 5.564.
>
> To show how training throughput scales with different sharing levels, we provide single-card training throughput (K tokens/second) and corresponding speedup:
>
> - *Original τ²-bench data (compression rate=9.427)*:
>
> | Method | Qwen3-1.7B | Qwen3-4B | Qwen3-8B | Qwen3-14B |
> | --- | --- | --- | --- | --- |
> | AReaL | 23.88K / 1.00x | 9.93K / 1.00x | 6.76K / 1.00x | 4.11K / 1.00x |
> | AReaL-DTA | 93.29K / 3.91x | 60.96K / 6.14x | 48.71K / 7.21x | 32.64K / 7.95x |
>
> - *Leaf-only τ²-bench data (compression rate=5.564)*:
>
> | Method | Qwen3-1.7B | Qwen3-4B | Qwen3-8B | Qwen3-14B |
> | --- | --- | --- | --- | --- |
> | AReaL | 23.83K / 1.00x | 9.95K / 1.00x | 6.77K / 1.00x | 4.17K / 1.00x |
> | AReaL-DTA | 56.62K / 2.38x | 36.79K / 3.70x | 29.24K / 4.32x | 19.53K / 4.68x |
>
> Furthermore, the dataset contains multiple batches with compression rates roughly distributed between 8 and 11 (for the original data). Examining the relationship between compression rate and speedup shows an approximately proportional correlation. The same conclusion holds when using leaf-only data. AReaL-DTA effectively exploits varying degrees of compression rates and does not rely solely on an extremely high compression rate. We agree that a motivating study would further strengthen the paper. We plan to extend our evaluation to additional agentic and long-horizon generation benchmarks in the updated manuscript to characterize how compression rates vary across tasks and how AReaL-DTA's speedup scales accordingly.

---

> > ### Author Rebuttal · Reviewer_b5PN · 2026-04-03
> >
> > Thanks for the response, which addressed my concerns. I will raise my rating.

---

> > > ### Author Response · Authors · 2026-04-03
> > >
> > > We sincerely appreciate your positive feedback and are glad that your concerns have been adequately addressed.

---

### Official Review · Reviewer_S6Do · 2026-03-17

**Soundness:** 3
**Presentation:** 3
**Significance:** 4
**Originality:** 4
**Overall Recommendation:** 5
**Confidence:** 3

**Summary:**

RL post-training for LLMs requires generating many rollout sequences per prompt. These sequences frequently share long common prefixes, yet existing systems compute those prefixes redundantly for every sequence. AREAL-DTA addresses this by organizing rollouts as a prefix tree and traversing it depth-first during both forward and backward passes. At any moment only one root-to-leaf path is held in memory — shared prefixes are computed once, the loss is computed and backpropagated immediately at each leaf, and activations are released before moving to the next branch. This keeps peak memory proportional to the longest sequence rather than the total token count across all sequences, eliminating the need for activation checkpointing. A companion load-balancing algorithm partitions sequences across GPUs in DFS order, preserving prefix sharing across workers while minimizing load imbalance. The method is implemented on top of the AReaL asynchronous RL framework and evaluated on τ²-bench with Qwen3 models ranging from 1.7B to 14B parameters, achieving up to 8.31× training throughput improvement over the baseline.

**Compliance With Llm Reviewing Policy:**

Affirmed.

**Key Questions For Authors:**

Are there algorithmic changes we can do when we have a lot of prefix-sharing trajectories? Can we estimate value in a different way than before? I understand that this paper only addresses the system efficiency, but would more performance be achieved through a better learning objective?

**Limitations:**

No limitations mentioned.

**Strengths And Weaknesses:**

## Strengths

Strong and well-decomposed efficiency gains. I really like the details provided in Section 3.2. the paper cleanly isolates the contribution of each optimization: basic tree traversal, leaf KV cache elimination, optimal DFS order, and chunked backpropagation. Each step is quantified across four model scales, making the engineering contributions transparent and reproducible.

## Weaknesses

Weaknesses

1. Only tested on τ²-bench. This benchmark has unusually long shared prefixes by nature. Whether the method helps on math or code RL tasks — the more common use cases — is never shown.
2. Only works when rollouts are similar. The efficiency gains disappear if rollouts diverge early. The paper never says when you should or shouldn't use this.

---

> ### Author Rebuttal · Authors · 2026-03-31
>
> > W2. Provide guidance to determine when DTA training helps.
>
> We appreciate this suggestion to discuss when DTA helps and will include this discussion in the updated manuscript. Concretely, we provide an analytical framework that quantitatively characterizes when DTA is beneficial, considering the simplified but representative setting in which all training sequences are leaf sequences. In DTA's DFS traversal, each backward pass processes one leaf sequence, and the effective per-pass token count will be $L_{\text{avg}} / C$, where $L_{\text{avg}}$ is the average leaf sequence length and $C$ is the compression rate (the ratio of tree tokens over the total tokens). The threshold condition for tree training to be more efficient than the vanilla baseline is when the computation w.r.t. $L_{\text{avg}} / C$ stays in the *compute-bound* regime of the GPU. For example, in terms of the Qwen3-14B, this threshold is approximately *768 tokens* on H200 GPUs, i.e., as long as each backward pass processes at least 768 tokens, the GPU's arithmetic cores will be highly utilized, and the throughput gain from prefix sharing will be reached without memory-bound degradation.
>
> > W1. Applicability beyond τ²-bench for coding and math benchmarks.
>
> Given the analytical framework to address W2, we present a concrete benefit analysis of τ²-bench and potential broader applicability.
>
> **τ²-bench analysis.** In our τ²-bench rollout data, the effective per-pass token count is approximately *1155 tokens* for leaf sequences, significantly exceeding the compute-bound threshold. Thus, τ²-bench achieves a high speedup for two reasons:
> - In multi-turn dialogues, thinking tokens from earlier turns are stripped in subsequent turns, so each leaf's backward pass covers at least the final-turn thinking and response tokens;
> - A common system-prompt prefix of approximately *4.5K tokens* is shared across all rollouts. These properties make τ²-bench a favorable yet *representative* case, where many real-world agentic tasks (e.g., tool-use agents, multi-step coding agents, a long global prompt) share a similar structure.
>
> **Broader applicability.** The prefix-sharing structure can still be favorable for many tasks. In fact, for math RL, if multi-turn reasoning is adopted, where the model iteratively refines its solution across turns, the same prefix-sharing dynamics will emerge: earlier reasoning turns become shared prefixes, and divergent suffixes are shortened by stripping intermediate thinking tokens. For code RL, tasks with long contexts (e.g., repository-level code generation or long-horizon debugging) naturally fall into the category of high $L_{\text{avg}}$, positioning $L_{\text{avg}} / C$ sufficiently above the compute-bound threshold even with moderate compression ratios. More generally, larger models have lower compute-bound thresholds due to higher arithmetic intensity, making tree training effective even with modest $C$. To make the experiments section more persuasive, we also plan to extend our evaluation to more math and code RL benchmarks in the revised manuscript.
>
>
>
> > Q1. Algorithmic opportunities from the prefix-sharing structure.
>
> This question raises an interesting discussion about a promising new research direction. In fact, we believe tree-structured rollouts open up algorithmic opportunities beyond system efficiency. At its core, the tree structure reflects an effort to maximize **per-token value**: each token's forward-pass hidden states are computed once yet contribute to multiple downstream sequences. This is purely system-level reuse in our current work, but the same structural insight suggests that the training algorithm itself could exploit more about efficient RL.
>
> Concretely, inspired by GRPO-style algorithms that contrast rewards across rollouts from the same prompt, we observe that the tree structure provides a more fine-grained signal. At each *fork point*, sibling branches diverge from identical hidden states and receive different eventual rewards. The reward difference between siblings, conditioned on the shared prefix, is a cleaner estimate of each branching decision's marginal value than a prompt-level baseline. Furthermore, the *entropy of the token distribution at the fork point* captures how uncertain the model was at the moment of divergence. Combining these two signals — reward contrast between sibling branches and fork-point entropy — could yield a learning objective that precisely targets the tokens where the model's decisions matter most. A fork with high entropy and large reward variance, for instance, indicates a decision point where the model is uncertain and the outcome is highly sensitive to the choice — a natural candidate for a stronger gradient signal.
>
> We believe designing tree-aware learning objectives that leverage the structural information available from tree-based rollouts is a compelling avenue for future work, and we plan to investigate it further as an independent follow-up work.

---

> > ### Author Rebuttal · Reviewer_S6Do · 2026-04-04
> >
> > The response is reasonable. I will keep my accept score.

---

> > > ### Author Response · Authors · 2026-04-08
> > >
> > > Thank you for your positive feedback. We’re pleased to hear that your concerns have been addressed.

---

### Official Review · Reviewer_K6hw · 2026-03-19

**Soundness:** 3
**Presentation:** 3
**Significance:** 3
**Originality:** 3
**Overall Recommendation:** 5
**Confidence:** 3

**Summary:**

This paper introduces "AREAL-DTA", an innovative framework designed to accelerate RL post-training of large language models. RL workflows often repeatedly recompute identical token prefixes across multiple generated sequences. The authors of the paper proposed an approach to efficiently exploit prefix sharing in RL training by organizing the rollouts into a prefix tree, and using a dynamic Depth-First Search traversal to compute shared tokens only once by interleaving forward and backward passes. Fo multi-GPU environments, the system utilizes a load-balanced partitioning strategy that sorts sequences by DFS order to minimize prefix duplication across workers. This helped the authors achieve higher training throughput and reduced peak GPU memory consumption.

**Compliance With Llm Reviewing Policy:**

Affirmed.

**Key Questions For Authors:**

1. In "Chunked back-propagation for long rollout sequences" section, the paper talks about an 'extra forward pass' needed for chunked back-propagation. I would like some more details on whether this ever causes any stability issues, and (or) any impact on speed.

**Limitations:**

yes

**Strengths And Weaknesses:**

**Soundness**
* Strength: The paper is technically sound. The DFS approach for gradient accumulation looks solid and ensures that all the shared prefix nodes receive gradient contributions from all their descendant leaves before they are popped. The space complexity analysis is clear and explains how  the peak memory usage is proportional to the length of the longest sequence.
* Weakness: In "Chunked back-propagation for long rollout sequences" section, the paper talks about an 'extra forward pass' needed for chunked back-propagation. I would like some more details on whether this ever causes any stability issues.

** Presentation**
The paper presentation is clear and Figure 1 in "introduction" section is helpful in visualizing how the stack states evolve during push and pop operations.

** Significance**
To accelerate RL post-training of large language models, the paper introduced a system that significantly increases RL training throughput and reduces memory consumption. This is relevant specially now when the community is looking for efficient RL training approaches.

** Originality**
Using dynamic DFS tree traversal in RL training with efficient memory usage is clever and original.

---

> ### Author Rebuttal · Authors · 2026-03-31
>
> >  W1, Q1. The impact of extra forward pass for chunked back-propagation in terms of  stability and speed.
>
> Thank you for this insightful question. We address the potential concern about the numerical stability and system efficiency of the extra forward pass below.
>
> **Numerical stability**: We conduct additional experiments to evaluate the numerical error of AReaL-DTA (with the extra forward pass and chunked back-propagation) against the original baseline on identical rollout data. And summarize the results below:
>
> - *FP32:* The maximal relative error for any aligned gradient value is within *2%* (i.e., some very unusual outlier), and the the median of the relative error for any aligned gradient value is less than *0.1%*, which indidates that for most of the positions, the error is negligible, confirming that the chunked recomputation introduces negligible numerical deviation in full precision.
> - *BF16:*  the maximal error of the end-to-end loss is also within *2%*, indicating that the training signal is well-preserved. An interesting observation is that the relative error of gradients is somehow more significant than that of FP32,  given BF16's limited mantissa bits, but the magnitude of the relative error of the gradients is comparable to the numerical error observed with flex attention under the same precision setting.
>
> Thus, we believe the extra forward pass does not introduce significant stability concerns beyond the inherent precision limitations of the chosen floating-point format. The error profile of AReaL-DTA matches with flex attention, a widely used tree-mask baseline, confirming that the chunked back-propagation strategy is numerically stable enough for RL training.
>
>
> **Impact for speed**: We conduct additional experiments to profile this overhead; the experimental results show that with `chunk_size = 4096`, the total time spent on extra forward passes accounts for *less than 10%* of the overall training time across all tested model sizes. Note that the reason we achieve improved efficiency is the *cut tail* optimization described in our paper, which avoids including unnecessary tokens at the tail of the tree during extra forward passes.

---

> > ### Author Rebuttal · Reviewer_K6hw · 2026-03-31
> >
> > Authors addressed the question asked in "Key Questions For Authors" section.

---

> > > ### Author Response · Authors · 2026-04-02
> > >
> > > Thank you for your positive feedback. We are glad that your concerns have been addressed.

---

### Decision · Program_Chairs · 2026-04-30

**Decision:**

Accept (regular)

**Comment:**

The paper proposes a method to optimize RL post-training of LLMs - by utilizing dynamic DFS traversal of rollout prefix trees to eliminate redundant computations and minimize memory footprint. This is a very interesting proposal and has high potential to significantly accelerate large-scale training efficiency. It's possible that the proposed work may not be applicable for tasks with low-prefix sharing. However, I still see substantial value for many practical tasks, particularly long-context reasoning tasks and also multi-step / multi-turn agentic tasks. I recommend accepting this paper.